# GANQ: GPU-Adaptive Non-Uniform Quantization for Large Language Models

**Pengxiang Zhao**[1]  **Xiaoming Yuan**[1]

## Abstract

Large Language Models (LLMs) face significant deployment challenges due to their substantial resource requirements. While low-bit quantized weights can reduce memory usage and improve inference efficiency, current hardware lacks native support for mixed-precision General Matrix Multiplication (mpGEMM), resulting in inefficient dequantization-based implementations. Moreover, uniform quantization methods often fail to capture weight distributions adequately, leading to performance degradation. We propose GANQ (GPU-Adaptive Non-Uniform Quantization), a layer-wise post-training non-uniform quantization framework optimized for hardware-efficient lookup table-based mpGEMM. GANQ achieves superior quantization performance by utilizing a training-free, GPU-adaptive optimization algorithm to efficiently reduce layer-wise quantization errors. Extensive experiments demonstrate GANQ's ability to reduce the perplexity gap from the FP16 baseline compared to state-of-the-art methods for both 3-bit and 4-bit quantization. Furthermore, when deployed on a single NVIDIA RTX 4090 GPU, GANQ's quantized models achieve up to $2.57\times$ speedup over the baseline, advancing memory and inference efficiency in LLM deployment.

## 1. Introduction

Large language models (LLMs) have demonstrated impressive performance across various domains (Brown et al., 2020; Achiam et al., 2023; Touvron et al., 2023a;b; Dubey et al., 2024; Gemini Team et al., 2023; Chowdhery et al., 2023; Zhang et al., 2023; Wang et al., 2023; Arefeen et al., 2024; Li et al., 2024a; Huang et al., 2024). However, their

deployment for inference remains challenging due to demanding resource requirements. For example, the LLaMA-3-70B (Dubey et al., 2024) model needs at least 140 GB of GPU memory in FP16, which exceeds current GPU capacities. While larger LLMs often yield better accuracy (Kaplan et al., 2020), these substantial resource demands hinder the practical deployment of LLMs, posing a barrier to their widespread adoption.

Quantization is a promising solution to reduce inference costs for LLMs. For example, 4-bit weight quantization can reduce memory usage for model loading by nearly 75% compared to FP16. In general, quantization techniques are categorized into quantization-aware training (QAT) and post-training quantization (PTQ). QAT integrates quantization into the training process to achieve higher accuracy but is computationally expensive, often requiring extensive samples and significant GPU hours (Liu et al., 2024). This makes QAT impractical for large models. In contrast, PTQ is a cost-effective alternative that applies quantization after training, making it the preferred choice for LLMs (Nagel et al., 2020; Yao et al., 2022; Frantar et al., 2022; Xiao et al., 2023; Dettmers et al., 2023; Kim et al., 2024; Lin et al., 2024; Shao et al., 2024; Ma et al., 2024; Li et al., 2024b). Among PTQ methods, weight-only quantization, which uses low-precision weights while retaining high-precision activations, has become a particularly attractive approach. By reducing memory traffic and alleviating memory-bound bottlenecks, weight-only quantization accelerates inference (Kim et al., 2024; Lin et al., 2024). Additionally, compared to weight-activation quantization, it avoids significant accuracy degradation by preserving the precision of activations, ensuring better model performance.

Despite its promise, weight-only quantization faces two key challenges. First, it shifts the core computation of LLM inference from standard General Matrix Multiplication (GEMM) to mixed-precision GEMM (mpGEMM), where low-precision weights (e.g., INT4/3/2) are multiplied with high-precision activations (e.g., FP16). Current hardware lacks native support for mpGEMM, necessitating dequantization to upscale low-bit weights into supported formats (see the left part of Figure 1(a)). This additional step introduces inefficiencies, particularly in large-batch scenarios, undermining the expected performance gains (Mo et al., 2024). Second, most existing methods rely on uni-

[1]Department of Mathematics, The University of Hong Kong, Hong Kong SAR, China. Correspondence to: Xiaoming Yuan <xmyuan@hku.hk>.

*Proceedings of the 42nd International Conference on Machine Learning*, Vancouver, Canada. PMLR 267, 2025. Copyright 2025 by the author(s).

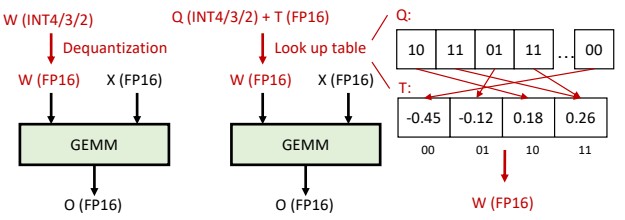
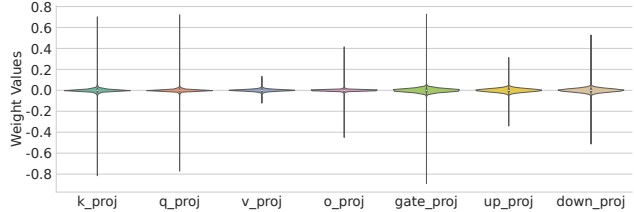

(a) Dequantization-based and LUT-based of mpGEMM.

(b) Violin plots of LLaMA-2-7B's first decoder layer weights.

*Figure 1.* (a) A comparison of two mpGEMM implementations: a dequantization-based approach (left) versus a LUT-based method (right). (b) Violin plots showing the first decoder layer's weight distribution in the LLaMA-2-7B model, clearly illustrating their deviation from a uniform distribution.

form quantization $\mathcal{Q} : \mathbb{R} \to [0, 2^N - 1] \cap \mathbb{Z}$ defined as $\mathcal{Q}(x) = \text{clamp}(\lfloor \frac{x}{s} \rceil + z, 0, 2^N - 1)$, where $\lfloor \cdot \rceil$ denotes rounding $N$ is the target bit width, $s$ is the scaling factor, and $z$ is the zero-point (Frantar et al., 2022; Xiao et al., 2023; Dettmers et al., 2023; Lin et al., 2024; Shao et al., 2024; Ma et al., 2024; Li et al., 2024b). However, LLM weight distributions are often highly non-uniform (see Figure 1(b)), making uniform quantization inadequate and resulting in suboptimal representations, particularly due to outliers. Techniques such as introducing learnable scale and zero-point parameters (Shao et al., 2024), applying affine transformations to preprocess weights (Ma et al., 2024), or splitting weights into various components and quantizing those that are easier to process (Dettmers et al., 2023; Li et al., 2024b), have been proposed to mitigate these issues. While these methods improve accuracy, they primarily address challenges within the uniform quantization framework rather than fundamentally enhancing the quantization method itself. Furthermore, they often increase computational complexity during inference due to the extra operations they require.

To address these issues, we propose GANQ (GPU-Adaptive Non-Uniform Quantization), a layer-wise post-training non-uniform quantization framework optimized for lookup table (LUT)-based mpGEMM. In LUT-based mpGEMM (see the right part of Figure 1(a)), complex computations are replaced with simple table lookups, supported by several GPU kernels (Kim et al., 2024; Mo et al., 2024; Guo et al., 2024). The primary challenge then becomes how to determine effective low-bit representations for the LUTs. Existing non-uniform quantization methods often rely on heuristic-based approaches, such as manually designed mappings (e.g., power-exponent functions (Yvinec et al., 2023)) or clustering-based methods with heuristic distance metrics (Han et al., 2015; Xu et al., 2018; Kim et al., 2024). While these methods may achieve good results in specific cases, their heuristic nature limits generalization and theoretical grounding. In contrast, GANQ introduces a principled optimization model for layer-wise LUT-based non-uniform quantization, formulated as a mixed-integer quadratic pro-

gramming problem. This model minimizes the discrepancy between the outputs of the quantized and original layers, thereby preserving accuracy. To efficiently address this complex model, GANQ utilizes its decomposable structure to divide the original optimization task into multiple independent one-dimensional subproblems, which can be processed in parallel using GPU acceleration to achieve substantial computational efficiency. Besides, GANQ employs an alternating direction optimization framework that capitalizes on the splittable structure of decision variables, effectively reducing quantization error.

In addition, although GANQ is designed as a base quantization method, it is fully compatible with current techniques for handling outliers, such as splitting weights into sparse components (to address outliers) and quantized components (Dettmers et al., 2023; Kim et al., 2024), thereby enabling further performance enhancements.

We evaluate GANQ extensively across various model families and sizes on language modeling tasks. The results show that GANQ consistently outperforms previous methods in quantization performance. Moreover, GANQ is highly resource-efficient and easy to implement. For instance, GANQ processes the LLaMA-2-7B model on a single NVIDIA RTX 4090 GPU in approximately one hour, using only 128 samples, each containing 2,048 tokens. Furthermore, our deployed models on the NVIDIA RTX 4090 GPU achieve up to $2.57\times$ speedups over the FP16 baseline by leveraging LUT-based inference kernels (Kim et al., 2024). These results highlight the effectiveness of GANQ in both quantization quality and inference efficiency.

## 2. Related Work

**Quantization for LLMs.** Quantization reduces the bit-precision of neural networks, resulting in smaller models and faster inference. It has become a key direction for compressing LLMs given their growing size and inference costs. Current quantization methods for LLMs are broadly categorized into QAT (Liu et al., 2024) and PTQ (Nagel

et al., 2020; Yao et al., 2022; Frantar et al., 2022; Xiao et al., 2023; Dettmers et al., 2023; Kim et al., 2024; Lin et al., 2024; Shao et al., 2024; Ma et al., 2024; Li et al., 2024b). QAT integrates quantization into the training process, preserving high performance but incurring prohibitive training costs, making it impractical for LLMs. In contrast, PTQ applies quantization to pretrained models, requiring only a small subset of data and modest computational resources, making it particularly appealing for LLMs. PTQ methods can be further classified into wight-only quantization and weight-activation quantization.

Weight-only quantization focuses on compressing model weights into low-bit formats. For example, GPTQ (Frantar et al., 2022) utilizes the optimal brain surgeon framework (Hassibi & Stork, 1992) for quantization and reconstruction. OmniQuant (Shao et al., 2024) introduces learnable parameters to determine quantization factors (e.g., scale and zero-point), while AffineQuant (Ma et al., 2024) extends this idea by incorporating a learnable matrix to preprocess weights before quantization. Weight-activation quantization compresses both weights and activations, often addressing their quantization jointly. For example, SmoothQuant (Xiao et al., 2023) shifts quantization difficulty from activations to weights using manually designed scaling factors. Similarly, SVDQuant (Li et al., 2024b) applies this approach while further decomposing weights into low-rank and quantized components.

While weight-activation quantization can offer broader compression, studies (Kim et al., 2024; Lin et al., 2024) have shown that LLM inference, especially during generation, is heavily memory-bound, with weight access dominating activation access by orders of magnitude. Consequently, weight-only quantization is more effective for on-device deployment of LLMs. In this work, we focus on weight-only PTQ for its efficiency and suitability for LLMs.

**Outlier Mitigation.** Due to the widely used uniform quantization mapping and the inherent non-uniform distribution of LLM weights, a key challenge is the presence of outliers. These outliers unnecessarily expand the quantization range (see Figure 1(b)), comprising quantization performance. Recent methods have been proposed to address this issue. For example, SpQR (Dettmers et al., 2023) and SqueezeLLM (Kim et al., 2024) retain outliers in sparse matrices while applying quantization to the remaining weights to mitigate their impact on overall performance. AWQ (Lin et al., 2024) independently quantizes the channel-wise salient weights to improve performance, and SVDQuant (Li et al., 2024b), as mentioned, decomposes weights into low-rank and quantized components. While these methods effectively handle outliers and enhance quantization performance, they often introduce additional computational overhead during inference. For instance, SpQR

and SqueezeLLM require both mpGEMM and sparse matrix multiplication, whereas SVDQuant adds an extra low-rank computation branch.

In this work, we propose a direct solution by introducing a non-uniform quantization framework that adapts to the distribution of LLM weights. Furthermore, our method is compatible with these outlier-handling techniques, enabling further performance enhancements when combined.

**LUT-based Inference Kernel.** Low-bit quantized LLMs depend on mpGEMM for efficient inference. This operation, which involves multiplying low-precision weights with higher-precision activations, presents a critical computational challenge. Current hardware lacks native support for mpGEMM, compelling existing implementations to adopt dequantization-based workarounds (Frantar et al., 2022; Lin et al., 2024; Shao et al., 2024; Ma et al., 2024; Li et al., 2024b). LUT-based methods offer a compelling alternative by eliminating dequantization overhead. Through efficient substitution of arithmetic operations with table lookups (Kim et al., 2024; Mo et al., 2024; Guo et al., 2024), these approaches demonstrate particular suitability for mpGEMM acceleration.

Unlike previous work focused on kernel-level optimizations, our research instead targets fundamental improvements in LUT-based quantization accuracy and compatibility with existing kernels.

**Non-Uniform Quantization.** The non-uniform distribution of weights in LLMs highlights the importance of non-uniform quantization. However, existing non-uniform quantization methods often rely on heuristic-based approaches, limiting their generalization and theoretical grounding. NU-PES (Yvinec et al., 2023) replaces uniform quantization with power-exponent functions and employs gradient-based optimization to learn the exponent parameter. Other methods focus on identifying shared weights, thereby forming a codebook, which is suitable for LUT-based mpGEMM. For example, Han et al. (2015) apply $k$-means clustering to minimize the Euclidean distance between weights and centroids in convolutional neural networks (CNNs), while Xu et al. (2018) extend this approach by using a loss-based metric for $k$-means clustering in CNNs. For LLMs, SqueezeLLM (Kim et al., 2024) adapts this idea by leveraging sensitivity-based $k$-means clustering, where the sensitivity metric measures the extent to which the model is perturbed after quantization. To mitigate the computational expense of this calculation, SqueezeLLM approximates the required Hessian matrix using the diagonal elements of the Fisher information matrix (Fisher, 1925).

In contrast, we propose a principled optimization model for layer-wise LUT-based non-uniform quantization for LLMs, along with an efficient GPU-adaptive algorithms to solve it.

# 3. Methodology

## 3.1. Optimization Model for Non-uniform Quantization

Consider a linear layer with weight matrix $\mathbf{W} \in \mathbb{R}^{m \times n}$ and input activation $\mathbf{X} \in \mathbb{R}^{n \times p}$, where $n$ represents the input hidden dimension, $m$ the output dimension, and $p = b \times s$ accounts for batched processing of $b$ sequences each of length $s$. As shown in the right part of Figure 1(a), LUT-based quantization aims to compress $\mathbf{W}$ by representing its elements using a codebook. Specifically, the elements of the $i$-th channel in the quantized weight matrix $\widetilde{\mathbf{W}}$ are selected from the codebook $\mathbf{T}_i = \{t_{i,0}, t_{i,1}, \ldots, t_{i,2^N-1}\}$, where $N$ is the bit-width of the quantization (e.g., 3 or 4 bits). Thus, each element $\widetilde{\mathbf{W}}_{i,j}$ satisfies $\widetilde{\mathbf{W}}_{i,j} \in \mathbf{T}_i$.

In practice, LUT-based quantization stores two components: a low-bit query matrix $\mathbf{Q} \in \{0, 1, \ldots 2^N - 1\}^{m \times n}$, which specifies the indices of values in the codebook, and the codebook itself, $\mathbf{T} \in \mathbb{R}^{m \times 2^N}$, which contains the quantized values for each channel. For example, if $\mathbf{Q}_{ij} = 0$, then $\widetilde{\mathbf{W}}_{i,j} = t_{i,0}$. Compared to the widely used basic per-channel uniform quantization (Frantar et al., 2022; Xiao et al., 2023), which requires two parameters per channel (i.e., scale and zero-point), this mechanism demands slightly more storage. However, as $\min\{m, n\} \gg 2^N$ in practice, the additional storage overhead is negligible. As shown in Table 1, for typical model sizes, the storage usage of LUT-based quantization remains comparable to the basic uniform quantization, differing by less than 0.2%. Moreover, some uniform quantization methods, such as OmniQuant (Shao et al., 2024) and AffineQuant (Ma et al., 2024), also require extra parameters.

To enable effective LUT-based non-uniform quantization, we formulate an optimization model aimed at minimizing the layer-wise output error.:

$$\min_{\mathbf{Q},\mathbf{T}} \|\mathbf{W}\mathbf{X} - \widetilde{\mathbf{W}}\mathbf{X}\|_F^2, \ s.t. \ \widetilde{\mathbf{W}}_{i,j} = \mathbf{T}_{i,\mathbf{Q}_{i,j}}, \ \forall i,j, \quad (1)$$

where $\| \cdot \|_F$ denotes the Frobenius norm, and $\mathbf{Q}$ and $\mathbf{T}$ are the decision variables.

Note that the quantized output for each row $(\widetilde{\mathbf{W}}\mathbf{X})_{i,:}$ depends only on its corresponding codebook $\mathbf{T}_{i,:}$ and query vector $\mathbf{Q}_{i,:}$. Consequently, the model in (1) is inherently decomposable across the rows of $\mathbf{W}$. Leveraging this property, the problem can be reformulated into $m$ independent subproblems, which are highly parallelizable and particularly suitable for GPU acceleration. Specifically, this parallelization is achieved by expressing computations in matrix form, which enables efficient matrix-vector and element-wise operations across rows. Furthermore, each subproblem can be expressed as a mixed-integer quadratic programming problem:

$$\min_{\mathbf{S}_i,\mathbf{T}_i} \|\mathbf{W}_i\mathbf{X} - \mathbf{T}_i\mathbf{S}_i\mathbf{X}\|^2 \ s.t. \ \mathbf{1}^\top\mathbf{S}_i = \mathbf{1}^\top, \ \forall i, \quad (2)$$

where $\mathbf{W}_i \in \mathbb{R}^{1 \times n}$ is the $i$-th row of $\mathbf{W}$, $\mathbf{T}_i \in \mathbb{R}^{1 \times 2^N}$ is the $i$-th row of $\mathbf{T}$, $\mathbf{S}_i \in \{0, 1\}^{2^N \times n}$ is a column-wise one-hot encoding matrix indicating the mapping of elements from $\mathbf{T}_i$, and $\mathbf{1}$ denotes an all-one vector.

The mixed-integer structure of $\mathbf{S}_i$ introduces significant combinatorial complexity, and the bilinear interaction between $\mathbf{S}_i$ and $\mathbf{T}_i$ in the objective further compounds the computational challenge, rendering the problem inherently non-convex and non-smooth. These factors pose serious difficulties for off-the-shelf solvers (Gurobi Optimization, 2025; IBM, 2025), especially in large-scale settings with high-dimensional weight matrices and input activations. In response, we develop a specialized, GPU-adaptive approach tailored to navigate this complex search space while scaling to practical problem sizes.

## 3.2. GPU-Adaptive Non-Uniform Quantization Method

To efficiently solve the model in (2) for LUT-based non-uniform quantization, we employ an alternating direction optimization framework. This framework iteratively updates $\mathbf{S}_i$ and $\mathbf{T}_i$ by decomposing the objective into two subproblems. Each subproblem optimizes one decision variable while keeping the other fixed. The iterative scheme is outlined as follows:

$$\begin{cases} \mathbf{S}_i^{k+1} = \underset{\mathbf{S}_i}{\operatorname{argmin}}\{\|\mathbf{W}_i\mathbf{X} - \mathbf{T}_i^k\mathbf{S}_i\mathbf{X}\|^2 \mid \mathbf{1}^\top\mathbf{S}_i = \mathbf{1}^\top\}, & (3) \\ \mathbf{T}_i^{k+1} = \underset{\mathbf{T}_i}{\operatorname{argmin}}\{\|\mathbf{W}_i\mathbf{X} - \mathbf{T}_i\mathbf{S}_i^{k+1}\mathbf{X}\|^2\}. & (4) \end{cases}$$

The $\mathbf{T}_i$-subproblem in (4) is an unconstrained quadratic program that admits a closed-form solution. Specifically, consider the objective function of $\mathbf{T}_i$-subproblem:

$$f(\mathbf{T}_i) = \|\mathbf{W}_i\mathbf{X} - \mathbf{T}_i\mathbf{S}_i^{k+1}\mathbf{X}\|^2. \quad (5)$$

To minimize this function, we apply the first-order optimality condition. Taking the gradient of $f(\mathbf{T}_i)$ with respect to $\mathbf{T}_i$ and setting it to zero yields:

$$\nabla f(\mathbf{T}_i) = 2(\mathbf{W}_i\mathbf{X} - \mathbf{T}_i\mathbf{S}_i^{k+1}\mathbf{X})\mathbf{X}^\top(\mathbf{S}_i^{k+1})^\top = \mathbf{0}. \quad (6)$$

Solving this matrix equation gives the closed-form update for $\mathbf{T}_i$:

$$\mathbf{T}_i^{k+1} = \mathbf{W}_i\mathbf{X}\mathbf{X}^\top(\mathbf{S}_i^{k+1})^\top((\mathbf{S}_i)^{k+1}\mathbf{X}\mathbf{X}^\top(\mathbf{S}_i^{k+1})^\top)^\dagger, \quad (7)$$

where $(\cdot)^\dagger$ denotes the Moore-Penrose inverse. Notably, the matrix $(\mathbf{S}_i)^{k+1}\mathbf{X}\mathbf{X}^\top(\mathbf{S}_i^{k+1})^\top$ has dimensions $2^N \times 2^N$, which is relatively small in practice (e.g., $16 \times 16$ under 4-bit quantization), ensuring that the computation remains efficient. Moreover, computing (7) involves only matrix-vector multiplications, making it highly efficient for GPU acceleration. Since the solutions to all $\mathbf{T}_i$-subproblems share the same formulation, they can be combined into a

*Table 1.* Storage requirements for full-precision (FP16), basic per-channel uniform quantization (4-bit), and per-channel LUT-based non-uniform quantization (4-bit) for weight matrix $\mathbf{W} \in \mathbb{R}^{m \times n}$. Percentages indicate storage usage relative to full-precision representation.

| CONFIGURATION | FULL (FP16) | BASIC UNIFORM (4-BIT) | LUT-BASED (4-BIT) |
|---|---|---|---|
| Theory | $2mn$ | $0.5mn + 4m$ | $0.5mn + 32m$ |
| $m = n = 2048$ (e.g., $\mathbf{W}_q$ in OPT-1.3B) | 100.00% | 25.10% | 25.78% |
| $m = n = 4096$ (e.g., $\mathbf{W}_q$ in LLaMA-2-7B) | 100.00% | 25.05% | 25.39% |
| $m = n = 8192$ (e.g., $\mathbf{W}_q$ in LLaMA-2-70B) | 100.00% | 25.02% | 25.20% |

single batch computation by stacking all $\mathbf{W}_i$ and $\mathbf{T}_i$ vectors row-wise and organizing $\mathbf{S}_i$ matrices into a tensor. Then, matrix operations can be used to efficiently compute the batch. This approach leverages modern GPUs' parallel processing capabilities, significantly reducing computational overhead and improving overall efficiency.

The primary challenge lies in the $\mathbf{S}_i$-subproblem (3), which is a discrete, non-convex, and non-smooth combinatorial optimization problem. In the case of 4-bit quantization, each element of $\mathbf{S}_i$ can assume one of 16 possible values. A brute-force search over all combinations would require $\mathcal{O}(16^n)$ operations, rendering it computationally prohibitive. Therefore, developing efficient solution techniques is essential for practical applications.

To address the $\mathbf{S}_i$-subproblem, we propose an efficient method that leverages the problem's inherent structure. The objective in (3) can be expanded as:

$$\|\mathbf{W}_i\mathbf{X} - \mathbf{T}_i^k\mathbf{S}_i\mathbf{X}\|^2 \tag{8}$$

$$= (\mathbf{W}_i - \mathbf{T}_i^k\mathbf{S}_i)(\mathbf{X}\mathbf{X}^\top)(\mathbf{W}_i - \mathbf{T}_i^k\mathbf{S}_i)^\top. \tag{9}$$

Then, consider the Cholesky decomposition of $\mathbf{X}\mathbf{X}^\top$:

$$\mathbf{X}\mathbf{X}^\top = \mathbf{L}\mathbf{L}^\top, \tag{10}$$

where $\mathbf{L}$ is a lower triangle matrix, meaning all its entries above the diagonal are zero.

*Remark* 3.1. If $\mathbf{X}\mathbf{X}^\top$ is not positive definite, which is rare but can occur in cases like the `fc2` layer of OPT models, we can add $\lambda\mathbf{I}$ ($\lambda > 0$) to guarantee positive definiteness before Cholesky decomposition. Specifically, for any non-zero vector $\mathbf{v}$, adjusted matrix satisfies $\mathbf{v}^\top(\mathbf{X}\mathbf{X}^\top + \lambda\mathbf{I})\mathbf{v} = \|\mathbf{X}^\top\mathbf{v}\|^2 + \lambda\|\mathbf{v}\|^2 > 0$. This preconditioning is a standard technique in numerical linear algebra.

Remark 3.1 describes a basic strategy to ensure positive definiteness by augmenting the matrix $\mathbf{X}\mathbf{X}^\top$ with a scaled identity matrix. However, selecting an appropriate $\lambda$ manually can be cumbersome and suboptimal. In practice, we adopt an adaptive preconditioning approach that enforces diagonal dominance, inherently ensuring positive definiteness without requiring manual hyperparameter tuning. Details are provided in Appendix A.

By combining (9) and (10), we have:

$$(\mathbf{W}_i - \mathbf{T}_i^k\mathbf{S}_i)(\mathbf{X}\mathbf{X}^\top)(\mathbf{W}_i - \mathbf{T}_i^k\mathbf{S}_i)^\top \tag{11}$$

$$= (\mathbf{W}_i - \mathbf{T}_i^k\mathbf{S}_i)(\mathbf{L}\mathbf{L}^\top)(\mathbf{W}_i - \mathbf{T}_i^k\mathbf{S}_i)^\top \tag{12}$$

$$= \|\mathbf{W}_i\mathbf{L} - \mathbf{T}_i^k\mathbf{S}_i\mathbf{L}\|^2. \tag{13}$$

Leverage the structure of $\mathbf{L}$, we minimize (13) using a back-substitution approach to efficiently derive a sub-optimal solution to (3). Specifically, there is

$$\|\mathbf{W}_i\mathbf{L} - \mathbf{T}_i^k\mathbf{S}_i\mathbf{L}\|^2 \tag{14}$$

$$= \sum_{j=0}^{n-1} \left( (\mathbf{W}_i\mathbf{L})_j - (\mathbf{T}_i^k\mathbf{S}_i\mathbf{L})_j \right)^2 \tag{15}$$

$$= \sum_{j=0}^{n-1} \left( \sum_{u=j}^{n-1} \left( \mathbf{W}_{i,u} - \mathbf{T}_i^k(\mathbf{S}_i)_{:,u} \right) \mathbf{L}_{u,j} \right)^2. \tag{16}$$

Following (16), we can solve for $\mathbf{S}_i$ from the last column ($j = n - 1$) to the first column ($j = 0$), minimizing each of the $n$ squared terms respectively. The $(n-1)$-th column of $\mathbf{L}$ has only one nonzero entry in rows $u \geq n - 1$, namely $\mathbf{L}_{n-1,n-1}$. Therefore, for $j = n - 1$, the residual involves a single term:

$$\left( \mathbf{W}_{i,n-1} - \mathbf{T}_i^k(\mathbf{S}_i)_{:,n-1} \right) \mathbf{L}_{n-1,n-1}. \tag{17}$$

Minimizing with respect to $(\mathbf{S}_i)_{:,n-1}$ gives that it should select an element from $\mathbf{T}_i^k$ that satisfies

$$\text{idx} = \underset{s}{\arg\min} \left| \mathbf{W}_{i,n-1} - \mathbf{T}_{i,s}^k \right|. \tag{18}$$

Then, we set $(\mathbf{S}_i)_{\text{idx},n-1} = 1$ and all other elements in this column to 0.

Once $(\mathbf{S}_i)_{:,n-1}$ is determined, the process moves to the $(n-2)$-th column. The residual becomes

$$\left( \mathbf{W}_{i,n-2} - \mathbf{T}_i^k(\mathbf{S}_i)_{:,n-2} \right) \mathbf{L}_{n-2,n-2} \tag{19}$$

$$+ \left( \mathbf{W}_{i,n-1} - \mathbf{T}_i^k(\mathbf{S}_i)_{:,n-1} \right) \mathbf{L}_{n-1,n-2}, \tag{20}$$

where (20) is a constant value given $(\mathbf{S}_i)_{:,n-1}$. In the following steps, we refer to $\mathbf{W}_{i,n-1} - \mathbf{T}_i^k(\mathbf{S}_i)_{:,n-1}$ as $r_{n-1}$. Then,

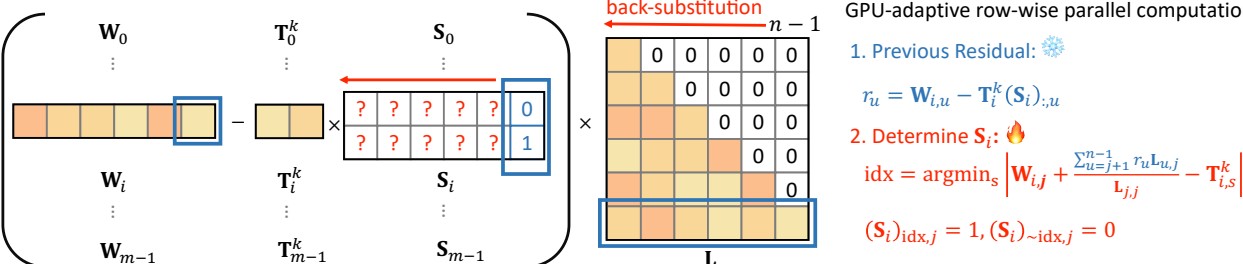

*Figure 2.* An illustration of the back-substitution framework for determining $\mathbf{S}_i$, leveraging the lower triangular structure of $\mathbf{L}$.

we solve for $(\mathbf{S}_i)_{:,n-2}$ by minimizing the square of (19) – (20):

$$\text{idx} = \underset{s}{\arg\min} \left| \mathbf{W}_{i,n-2} + \frac{r_{n-1}\mathbf{L}_{n-1,n-2}}{\mathbf{L}_{n-2,n-2}} - \mathbf{T}_{i,s}^k \right|, \quad (21)$$

and we set $(\mathbf{S}_i)_{\text{idx},n-2} = 1$ and the rest of $(\mathbf{S}_i)_{:,n-2} = 0$.

This back-substitution process continues for $j = n - 3, \ldots, 0$. At each step the element of $(\mathbf{S}_i)_{\text{idx},j}$ set to 1 is determined as

$$\text{idx} = \underset{s}{\arg\min} \left| \mathbf{W}_{i,j} + \frac{1}{\mathbf{L}_{j,j}} \sum_{u=j+1}^{n-1} r_u\mathbf{L}_{u,j} - \mathbf{T}_{i,s}^k \right|. \quad (22)$$

where $r_u = \mathbf{W}_{i,u} - \mathbf{T}_i^k(\mathbf{S}_i)_{:,u}$.

Figure 2 illustrates the back-substitution framework for efficiently determining $\mathbf{S}_i$. Since the solution processes for $\mathbf{S}_i, i = 0, 1, \ldots, m - 1$ are independent, similar to the batch solving of $\mathbf{T}_i$-subproblems described earlier, we can stack all $\mathbf{W}_i$ and $\mathbf{T}_i$ vectors row-wise and organize the $\mathbf{S}_i$ matrices into a tensor. This allows the back-substitution process to be performed for the entire problem using matrix operations, leveraging modern GPUs' parallel processing capabilities to enhance overall efficiency.

Finally, the full pseudocode of GANQ for layer-wise LUT-based non-uniform quantization is presented in Algorithm 1.

### 3.3. Compatibility with Outlier-Handling Techniques

GANQ provides a foundational framework for LUT-based non-uniform quantization and is inherently compatible with existing techniques for handling outliers in weight matrices. Among these techniques, a widely adopted approach involves splitting the weight matrix into a sparse matrix for outliers and a quantized matrix for the remaining weights. For example, SpQR (Dettmers et al., 2023) and SqueezeLLM (Kim et al., 2024) extract outliers into a separate sparse matrix to mitigate their impact on the quantization process.

---

**Algorithm 1** GANQ: GPU-Adaptive Layer-Wise LUT-Based Non-Uniform Quantization

**Input:** $\mathbf{W} \in \mathbb{R}^{m \times n}$, $\mathbf{X} \in \mathbb{R}^{n \times p}$, initial codebook $\mathbf{T}^0 \in \mathbb{R}^{m \times 2^N}$, number of iterations $K$
**Output:** Updated $\mathbf{T}^K$ and query matrix $\mathbf{Q}^K \in \{0, 2^N - 1\}^{m \times n}$
Initialize $\mathbf{S}^0 = \mathbf{0}^{m \times 2^N \times n}$                    # tensor format
Compute $\mathbf{H} = \mathbf{X}\mathbf{X}^\top$
Compute $\mathbf{L} = \text{Cholesky}(\mathbf{H})$        # Cholesky decomposition
**for** $k \leftarrow 0$ to $K - 1$ **do**
    Initialize $\mathbf{r} = \mathbf{0}^{m \times 1}$              # previous residual vector
    **for** $j \leftarrow n - 1$ to $0$ **do**
        $\text{idx} = \arg\min_\mathbf{s} \left| \mathbf{W}_{:,j} + \frac{\mathbf{r}}{\mathbf{L}_{j,j}} - \mathbf{T}_{:,\mathbf{s}}^k \right|$        # row-wise
        $\mathbf{Q}_{:,j}^{k+1} = \text{idx}$
        Update $\mathbf{S}_{:,:,j}^{k+1}$ using $\text{idx}$        # one-hot encoding
        $\mathbf{r} = (\mathbf{W}_{:,j:} - \mathbf{T}^k\mathbf{S}_{:,:,j:}^{k+1})\mathbf{L}_{j:,j-1}$        # update residual
    **end for**
    $\mathbf{T}^{k+1} = \mathbf{W}\mathbf{H}(\mathbf{S}^{k+1})^\top((\mathbf{S}^{k+1})\mathbf{H}^\top(\mathbf{S}^{k+1})^\top)^\dagger$ # batch update
**end for**
**Return** $\mathbf{T}^K, \mathbf{Q}^K$

---

In our framework, the weight matrix $\mathbf{W}$ can similarly be decomposed into a sparse component $\mathbf{W}_{\text{sparse}}$, containing extracted outliers, and a dense component $\mathbf{W}_{\text{dense}}$, processed through GANQ. Appendix B details our outlier extraction method, which identifies outliers using a small ratio threshold $r$ (e.g., $r = 0.5\%$ of total parameters) while preserving the remaining weights for quantization. This decomposition reduces quantization range, thereby enhancing the quantization performance.

## 4. Experiments

### 4.1. Settings

**Quantization.** We evaluate GANQ on weight-only non-uniform quantization. The default configuration employs INT4/3 per-channel weight quantization.

**Models.** We comprehensively evaluate GANQ on a range of models, including OPT (Zhang et al., 2022), LLaMA (Touvron et al., 2023a), LLaMA-2 (Touvron et al., 2023b), LLaMA-3 (Meta AI, 2024a), and LLaMA-3.2 (Meta AI,

2024b) model families. Specifically, we assess its performance across OPT-125M, OPT-350M, OPT-1.3B, OPT-2.7B, OPT-6.7B, LLaMA-7B, LLaMA-2-7B, LLaMA-3-8B, LLaMA-3.2-1B, LLaMA-3.2-3B, LLaMA-3.2-1B-Instruct, and LLaMA-3.2-3B-Instruct models.

**Evaluation.** Following prior work (Frantar et al., 2022; Shao et al., 2024; Ma et al., 2024; Kim et al., 2024), we evaluate the quantized models by reporting perplexity on language datasets, specifically using the WikiText-2 (Merity et al., 2017), C4 (Raffel et al., 2020), and PTB (Marcus et al., 1994) datasets. Consistent with established practice, we use a sequence length of 2,048 across all models. Additionally, we assess accuracy on zero-shot tasks, including ARC Easy, ARC Challenge (Clark et al., 2018), WinoGrande (Sakaguchi et al., 2021), BoolQ (Clark et al., 2019), RTE (Wang et al., 2018), HellaSwag (Zellers et al., 2019), and GSM8K (Cobbe et al., 2021), facilitated by the LM Harness library (Gao et al., 2021). Long-context capabilities are evaluated using LongBench (Bai et al., 2024) under its standard protocol.

**Baselines.** For basic weight-only quantization, we compare GANQ with standard round-to-nearest uniform quantization (RTN), GPTQ (Frantar et al., 2022), and OmniQuant (Shao et al., 2024). For weight-only quantization with outlier handling, we compare with GPTQ, OmniQuant, and AWQ (Lin et al., 2024), each using a group size of 128, as well as SqueezeLLM (Kim et al., 2024).

**Setup.** We implement GANQ using the PyTorch (Paszke et al., 2019) and utilize the HuggingFace Transformers library (Wolf, 2019) for model and dataset management. Our implementation is publicly available[1]. All experiments are conducted on a single NVIDIA RTX 4090 GPU. For calibration data, we follow the methodology outlined in previous works (Frantar et al., 2022; Shao et al., 2024; Kim et al., 2024). Specifically, we use 32 sequences for OPT models and 128 sequences for LLaMA models. Each sequence consists of 2,048 tokens, sampled from the first shard of the C4 dataset.

**Latency Profiling.** Using Torch CUDA profiler, we measure single-sequence (batch size 1) generation of 1024 tokens on a single NVIDIA RTX 4090 GPU, reporting CUDA time and peak memory usage with LUT-based inference kernels in (Kim et al., 2024).

### 4.2. Main Results

**Weight-only Quantization.** The results in Table 2 present the WikiText-2 perplexity of quantized OPT, LLaMA, LLaMA-2, and LLaMA-3 models under 4-bit and 3-bit configurations across different model sizes (with additional

---

[1]The code is available at https://github.com/Evans-Z/GANQ

perplexity results on the C4 and PTB datasets as well as results of quantized LLaMA-3.2 models in Appendix C). As shown, GANQ consistently outperforms baseline methods such as RTN, GPTQ, and OmniQuant across all configurations. For 4-bit quantization, GANQ achieves the lowest perplexity across both OPT and LLaMA models, with notable improvements. Remarkably, on OPT-2.7B, GANQ's perplexity (12.33) even outperforms the full-precision FP16 model (12.47). GANQ also demonstrates strong performance with 3-bit quantization, maintaining competitive perplexity reductions across model sizes. For example, on OPT-6.7B, GANQ's perplexity is 11.39, compared to 15.11 for GPTQ and 13.47 for OmniQuant. These results underscore GANQ's effectiveness in both 4-bit and 3-bit quantization, achieving substantial perplexity reductions across various model scales. The "–" in Table 2 indicates that OmniQuant cannot quantize LLaMA-3-8B on a single NVIDIA RTX 4090 GPU due to memory constraints or the unavailability of the pre-quantized model.

The results in Table 3 show the zero-shot performance of the quantized LLaMA-2-7B model across six tasks under 4-bit and 3-bit quantization. GANQ outperforms baseline methods such as RTN, GPTQ, and OmniQuant in both bit-width configurations. With 4-bit quantization, GANQ achieves an average accuracy of 64.23%, which is comparable to the full-precision model (64.47%). For 3-bit quantization, GANQ maintains strong performance with an average accuracy of 62.22%, significantly surpassing other baseline methods. These results demonstrate GANQ's ability to preserve high task performance, even under aggressive quantization.

Table 4 presents a performance comparison of 4-bit quantized LLaMA-3.2 1B-Instruct and 3B-Instruct models on two tasks: LongBench, which evaluates long-context capabilities, and GSM8K, which assesses reasoning-intensive tasks. Both tasks are evaluated in zero-shot settings without chain-of-thought prompting. GANQ consistently outperforms baseline methods on both tasks, highlighting its robustness. In contrast, RTN exhibits extreme sensitivity to model scale, collapsing on LLaMA-3.2-1B-Instruct. Similarly, GPTQ achieves near-zero GSM8K accuracy on LLaMA-3.2-3B-Instruct, consistent with the perplexity results reported in Table 10, and encounters LongBench execution errors (i.e., `skip_special_tokens` failures) despite identical experimental setups. Compared to RTN and GPTQ, OmniQuant leverages learnable quantization parameters, leading to relatively stable performance. However, it still falls short of GANQ's results. These findings underscore GANQ's ability to deliver superior performance across diverse tasks.

**Weight-only Quantization with Outlier Handling.** To mitigate outlier impact, methods like RTN, GPTQ, AWQ, and OmniQuant divide per-channel distributions into smaller

*Table 2.* WikiText-2 perplexity (↓) of quantized models under 4-bit and 3-bit. GANQ outperforms state-of-the-art methods.

| Method | Bit-width | OPT | | | | | LLaMA | | |
|---|---|---|---|---|---|---|---|---|---|
| | | 125M | 350M | 1.3B | 2.7B | 6.7B | 7B | 2-7B | 3-8B |
| Full | 16 | 27.66 | 22.00 | 14.63 | 12.47 | 10.86 | 5.68 | 5.47 | 6.13 |
| RTN | 4 | 37.11 | 25.94 | 48.18 | 16.73 | 12.14 | 6.29 | 6.12 | 8.53 |
| GPTQ | 4 | 31.08 | 23.99 | 15.60 | 12.88 | 11.45 | 6.95 | 6.08 | 2.4e2 |
| OmniQuant | 4 | 30.98 | 23.34 | 15.25 | 12.84 | 11.25 | 5.92 | 5.88 | – |
| GANQ | 4 | **28.58** | **23.04** | **14.94** | **12.33** | **10.70** | **5.83** | **5.65** | **6.61** |
| RTN | 3 | 1.3e3 | 64.56 | 1.3e4 | 1.3e4 | 5.8e3 | 25.54 | 5.4e2 | 2.2e3 |
| GPTQ | 3 | 52.48 | 34.47 | 21.60 | 16.95 | 15.11 | 16.65 | 9.46 | 1.4e2 |
| OmniQuant | 3 | 42.43 | 29.64 | 18.22 | 19.47 | 13.47 | 6.79 | 7.08 | – |
| GANQ | 3 | **35.98** | **29.42** | **17.05** | **14.10** | **11.39** | **6.33** | **6.25** | **8.83** |

*Table 3.* Accuracies (%, ↑) of the quantized LLaMA-2-7B model on 6 zero-shot tasks under 4-bit and 3-bit quantization.

| Method | Bit-width | HellaSwag | BoolQ | RTE | WinoGrande | Arc-e | Arc-c | Mean |
|---|---|---|---|---|---|---|---|---|
| Full | 16 | 57.12 | 77.68 | 63.18 | 69.06 | 76.35 | 43.43 | 64.47 |
| RTN | 4 | 55.59 | 73.61 | 59.57 | 68.43 | 74.03 | 41.30 | 62.09 |
| GPTQ | 4 | 55.66 | 74.43 | 57.76 | 57.72 | **75.25** | 42.32 | 60.52 |
| OmniQuant | 4 | 55.66 | 75.69 | 63.90 | 68.19 | 74.33 | 39.85 | 62.94 |
| GANQ | 4 | **56.10** | **77.31** | **65.70** | **68.75** | 74.96 | **42.58** | **64.23** |
| RTN | 3 | 30.93 | 42.54 | 52.71 | 52.17 | 34.76 | 21.33 | 39.07 |
| GPTQ | 3 | 47.55 | 67.03 | 55.60 | 59.75 | 64.60 | 33.11 | 54.61 |
| OmniQuant | 3 | 52.58 | 72.11 | 57.40 | 64.72 | 68.73 | 36.43 | 58.66 |
| GANQ | 3 | **53.85** | **75.02** | **62.82** | **67.48** | **73.36** | **40.78** | **62.22** |

*Table 4.* Performance comparison of 4-bit quantized LLaMA-3.2 1B-Instruct and 3B-Instruct models on LongBench (↑) and GSM8K (↑) datasets.

| Method | 1B-Instruct | | 3B-Instruct | |
|---|---|---|---|---|
| | LongBench | GSM8K (%) | LongBench | GSM8K (%) |
| FP16 | 11.5 | 32.90 | 12.7 | 64.97 |
| RTN | 0.2 | 4.17 | 12.1 | 37.68 |
| GPTQ | – | 11.14 | – | – |
| OmniQuant | 8.9 | 16.53 | 11.5 | 54.74 |
| GANQ | **11.7** | **27.75** | **12.5** | **60.50** |

blocks (typically of size 128). SqueezeLLM retains a small percentage of outliers (e.g., 0.5%) and a fixed number of full rows (default: 10). GANQ can integrate seamlessly with SqueezeLLM's outlier handling mechanism. We evaluate this integration through experiments. Due to memory constraints, OmniQuant cannot quantize LLaMA-3-8B, and SqueezeLLM is limited to models up to 2.7B. We use results directly from their paper for these cases if available. Additionally, SqueezeLLM's current code does not support OPT-350M. For GANQ, we retain 0.5% outliers for all OPT models and LLaMA-3-8B. Additionally, we retain 10 full rows for LLaMA-7B and LLaMA-2-7B to ensure a fair comparison with SqueezeLLM.

As shown in Table 5, GANQ* (indicating GANQ integrated with outlier handling) outperforms other baselines. Furthermore, when retaining only 0.5% of outliers for LLaMA-7B and LLaMA-2-7B, we observe the following results: LLaMA-7B (5.78 for 4-bit, 6.20 for 3-bit), LLaMA-2-7B (5.60 for 4-bit, 6.10 for 3-bit), which still outperform all other methods except for SqueezeLLM.

## 4.3. Profiling

Table 6 reports CUDA time, speedup, and peak memory usage for GANQ and GANQ*, measured on a single NVIDIA RTX 4090 GPU while generating 1,024 tokens with a batch size of 1 on OPT-6.7B and LLaMA-7B. GANQ achieves up to a 2.57× speedup over the FP16 baseline, with peak memory usage reduced to 4.10 GB for OPT-6.7B and 3.30 GB for LLaMA-7B under 3-bit quantization. While GANQ* provides improved quantization accuracy through outlier handling, it incurs slightly higher inference latency due to additional sparse matrix operations. Therefore, the choice between GANQ and GANQ* should be guided by the desired trade-off between accuracy and runtime efficiency.

The above results demonstrate that GANQ is fully compatible with existing LUT-based inference kernels (Kim et al., 2024). Moreover, GANQ stands to benefit from ongoing engineering advancements in this class of kernels, such as

*Table 5.* WikiText-2 perplexity (↓) of quantized models under 4-bit and 3-bit. GANQ outperforms state-of-the-art methods.

| Method | Bit-width | OPT | | | | | LLaMA | | |
|---|---|---|---|---|---|---|---|---|---|
| | | 125M | 350M | 1.3B | 2.7B | 6.7B | 7B | 2-7B | 3-8B |
| Full | 16 | 27.66 | 22.00 | 14.63 | 12.47 | 10.86 | 5.68 | 5.47 | 6.13 |
| RTN (g128) | 4 | 30.49 | 24.51 | 15.29 | 12.80 | 11.15 | 5.96 | 5.72 | 6.73 |
| GPTQ (g128) | 4 | 29.78 | 23.40 | 14.91 | 12.50 | 10.99 | 6.40 | 5.65 | 8.97 |
| AWQ (g128) | 4 | 29.09 | 1.3e4 | 14.93 | 12.70 | 10.96 | 5.78 | 5.60 | 6.53 |
| OmniQuant (g128) | 4 | 29.57 | 22.85 | 14.88 | 12.66 | 11.04 | 5.79 | 5.62 | – |
| SqueezeLLM | 4 | 28.51 | – | 14.83 | 12.60 | 10.92 | 5.77 | 5.57 | – |
| GANQ$^\star$ | 4 | **28.16** | **22.84** | **14.53** | **12.19** | **10.69** | **5.76** | **5.57** | **6.50** |
| RTN (g128) | 3 | 50.61 | 36.33 | 1.2e2 | 2.6e2 | 22.87 | 7.01 | 6.66 | 12.07 |
| GPTQ (g128) | 3 | 37.93 | 28.21 | 16.33 | 13.57 | 11.30 | 8.68 | 6.43 | 19.87 |
| AWQ (g128) | 3 | 35.75 | 1.7e4 | 16.31 | 13.56 | 11.39 | 6.35 | 6.24 | 8.22 |
| OmniQuant (g128) | 3 | 35.61 | 27.65 | 16.16 | 13.28 | 11.23 | 6.30 | 6.23 | – |
| SqueezeLLM | 3 | 32.59 | – | 15.76 | 13.43 | 11.31 | 6.13 | 5.96 | – |
| GANQ$^\star$ | 3 | **32.35** | **26.84** | **15.52** | **13.11** | **11.13** | **6.08** | **5.93** | **7.46** |

*Table 6.* CUDA time (s), speedup (↑), and peak memory (GB) (↓) for GANQ and GANQ$^\star$ using the LUT-based inference kernel from (Kim et al., 2024) on OPT-6.7B and LLaMA-7B models.

| Method | Bit-width | OPT-6.7B | | | LLaMA-7B | | |
|---|---|---|---|---|---|---|---|
| | | CUDA time | Speedup (↑) | Peak Memory (↓) | CUDA time | Speedup (↑) | Peak Memory (↓) |
| Full | 16 | 16.76 | 1.0 | 12.91 | 17.86 | 1.00 | 13.06 |
| GANQ | 4 | 7.47 | 2.24 | 4.88 | 8.46 | 2.11 | 4.14 |
| GANQ$^\star$ | 4 | 10.39 | 1.61 | 5.13 | 9.82 | 1.82 | 4.12 |
| GANQ | 3 | 6.51 | 2.57 | 4.10 | 7.48 | 2.39 | 3.30 |
| GANQ$^\star$ | 3 | 10.73 | 1.56 | 4.39 | 8.85 | 2.02 | 3.32 |

those proposed in (Guo et al., 2024; Mo et al., 2024), which can further enhance implementation efficiency and overall inference performance.

### 4.4. Quantization Cost

Among the evaluated methods, RTN, GPTQ, and AWQ are the most efficient in GPU memory usage and quantization time, due to their layer-wise heuristic approach. However, they trade off model performance. In contrast, OmniQuant and SqueezeLLM require gradient information, leading to higher memory demands. OmniQuant can quantize 7B models on a single NVIDIA RTX 4090 GPU but fails for 8B models and takes over 3 hours for 7B quantization. SqueezeLLM, which requires global gradients, can only quantize models up to 2.7B. GANQ leverages GPU-adaptive, parallel row-wise computation to quantize 7B models in approximately 1 hour for $K = 10$. Considering both model performance and quantization cost, GANQ is an effective solution.

### 5. Conclusion

In this work, we introduce GANQ, a GPU-adaptive non-uniform quantization framework for efficient deployment and inference of LLMs. GANQ introduces a principled optimization model for layer-wise LUT-based quantization, formulated as a mixed-integer quadratic programming problem, and solves it efficiently using a GPU-adaptive alternating direction optimization algorithm. Extensive experiments demonstrate that GANQ reduces perplexity compared to state-of-the-art methods in 3-bit and 4-bit quantization settings, while preserving high accuracy. Additionally, GANQ achieves up to $2.57\times$ speedup over FP16 baselines on a single NVIDIA RTX 4090 GPU, showcasing its substantial improvements in both memory usage and inference efficiency. GANQ is highly resource-efficient, easy to implement, and compatible with existing techniques for handling outliers, making it a highly flexible solution for large-scale LLM deployment. These results highlight GANQ's potential to enable practical and efficient deployment of high-performance LLMs across a wide range of hardware configurations.

## Acknowledgments

The work of X. Yuan was supported by the GRF RGC (No. 17309824).

## Impact Statement

This paper presents work whose goal is to advance the field of Machine Learning. There are many potential societal consequences of our work, none which we feel must be specifically highlighted here.

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

## A. Ensuring Positive Definiteness via Diagonal Dominance

In some cases, such as the `fc2` layer of OPT models, the matrix $\mathbf{XX}^\top$ may fail to be positive definite. While this is uncommon in our experiments, it is important to handle such situations before performing Cholesky decomposition in Algorithm 1, which requires a positive definite matrix. To ensure this, we enforce diagonal dominance via a small adjustment to the matrix.

Specifically, a symmetric matrix $\mathbf{A}$ is guaranteed to be positive definite if it is diagonally dominant with strictly positive diagonal entries, meaning $|a_{ii}| \geq \sum_{j \neq i} |a_{ij}|$ for all $i$. To utilize this property, let us denote $\mathbf{\Sigma} = \mathbf{XX}^\top$, where diagonal elements satisfy $\mathbf{\Sigma}_{ii} \geq 0$. We compute an adaptive offset vector $\boldsymbol{\delta}$, where each component $\boldsymbol{\delta}_i$ is calculated as:

$$\boldsymbol{\delta}_i = \max\left(\sum_{j=1}^n |\mathbf{\Sigma}_{i,j}| - 2\mathbf{\Sigma}_{i,i},\ 10^{-8}\right), \tag{23}$$

where $\mathbf{\Sigma}_{i,j}$ denotes the element at the $i$-th row and $j$-th column of $\mathbf{\Sigma}$. Subsequently, we obtain a diagonally dominant matrix by adding this adaptive diagonal offset and perform the Cholesky decomposition:

$$\mathbf{L} = \text{Cholesky}\left(\mathbf{\Sigma} + \text{Diag}(\boldsymbol{\delta})\right), \tag{24}$$

where $\text{Diag}(\boldsymbol{\delta})$ generates a diagonal matrix from the vector $\boldsymbol{\delta}$, and $\mathbf{L}$ is the resulting lower-triangular Cholesky factor. This adaptive preconditioning procedure efficiently ensures numerical stability and positive definiteness, facilitating robust computation without manual parameter adjustments.

To evaluate sensitivity to preconditioning approaches (the standard fixed $\lambda$ illustrated in Remark 3.1 versus our adaptive method), we conduct 4-bit quantization experiments on OPT-125M and report the resulting perplexity on WikiText-2 in Table 7.

*Table 7.* WikiText-2 perplexity ($\downarrow$) of 4-bit quantized OPT-125M under different preconditioning strategies.

| $\mathbf{XX}^\top + \lambda\mathbf{I}$ | | | | | Method in (23) – (24) |
|---|---|---|---|---|---|
| $\lambda = 0.5$ | $\lambda = 1.0$ | $\lambda = 10.0$ | $\lambda = 40.0$ | $\lambda = 100.0$ | |
| 29.14 | 29.04 | 28.98 | 29.05 | 29.09 | 28.58 |

The results show that quantization accuracy of GANQ is largely insensitive to the choice of preconditioning. The adaptive method achieve the best result, with fixed $\lambda$ yielding similar results. All results outperforms baselines, confirming the robustness.

## B. Outlier Extraction Method

In this section, we describe the method we implemented to extract outliers from the matrix $\mathbf{W}$. This method decomposes $\mathbf{W}$ into two components: a sparse component $\mathbf{W}_{\text{sparse}}$, which contains the extracted outliers, and a dense component $\mathbf{W}_{\text{dense}}$, which consists of the remaining non-outlier values. We then perform quantization on $\mathbf{W}_{\text{dense}}$ to enhance quantization performance.

The pseudocode is shown in Algorithm 2. This decomposition is achieved through a row-wise outlier extraction process based on an extraction ratio $r$, where $0 < r < 1$ (e.g., 0.5%). The process begins by computing a tail percentile $p = 1 - 0.5 \times r$, which determines the boundaries for identifying outliers in each row symmetrically. For each row of the weight matrix, the algorithm sorts the values in ascending order and computes the upper and lower cutoff values corresponding to the percentiles $p$ and $1 - p$. These cutoff values define the outliers, which are those values that fall either above the upper percentile or below the lower percentile. An outlier mask $\mathbf{M}$ is then created, where values that are identified as outliers are marked with 1, and non-outliers are marked with 0. The sparse component $\mathbf{W}_{\text{sparse}}$ is obtained by multiplying the weight matrix $\mathbf{W}$ element-wise with the outlier mask, while the dense component $\mathbf{W}_{\text{dense}}$ is obtained by subtracting the sparse component from the original matrix.

---

**Algorithm 2** Outlier Extraction and Weight Decomposition

---

**Input:** Weight matrix $\mathbf{W} \in \mathbb{R}^{m \times n}$, outlier extraction ratio $0 < r < 1$
**Output:** Sparse component $\mathbf{W}_{\text{sparse}}$, Dense component $\mathbf{W}_{\text{dense}}$
$p \leftarrow 1 - 0.5 \times r$                                                     # compute tail percentile
$\mathbf{M} \leftarrow \mathbf{0}$                                                   # initialize outlier mask $\mathbf{M}$ with zeros
$\text{upper} \leftarrow \lfloor n \times p \rfloor$
$\mathbf{W}_{\text{sorted}} \leftarrow \text{sort}(\mathbf{W}[:])$                   # row-wise sorting
$\mathbf{c}_{\text{upper}} \leftarrow \mathbf{W}_{\text{sorted}}[:, \text{upper}]$   # upper cutoff values
$\text{lower} \leftarrow \lceil n \times (1 - p) \rceil$
$\mathbf{c}_{\text{lower}} \leftarrow \mathbf{W}_{\text{sorted}}[:, \text{lower}]$   # lower cutoff values
$\mathbf{O} \leftarrow (\mathbf{W} \geq \mathbf{c}_{\text{upper}}) \vee (\mathbf{W} \leq \mathbf{c}_{\text{lower}})$   # identify outliers
$\mathbf{M}[\mathbf{O}] \leftarrow 1$                                                # mark outliers in the mask
$\mathbf{W}_{\text{sparse}} \leftarrow \mathbf{W} \circ \mathbf{M}$                  # extract outliers
$\mathbf{W}_{\text{dense}} \leftarrow \mathbf{W} - \mathbf{W}_{\text{sparse}}$       # extract non-outliers
**Return** $\mathbf{W}_{\text{sparse}}, \mathbf{W}_{\text{dense}}$

---

# C. Additional Results

The results in Table 8 present the C4 perplexity of various quantized models under 4-bit and 3-bit configurations across different model sizes. As shown, GANQ outperforms baseline methods such as RTN, GPTQ, and OmniQuant across all configurations. For 4-bit quantization, GANQ achieves the lowest perplexity across both OPT and LLaMA models, with notable improvements. GANQ also demonstrates strong performance with 3-bit quantization, maintaining competitive perplexity reductions across model sizes. On OPT-6.7B, GANQ's perplexity is 13.68, compared to 17.00 for GPTQ and 15.51 for OmniQuant. These results highlight GANQ's effectiveness in both 4-bit and 3-bit quantization, achieving substantial perplexity reductions across a wide range of model scales. The symbol "–" in Table 8 indicates that OmniQuant cannot quantize LLaMA-3-8B on a single NVIDIA RTX 4090 GPU due to memory constraints, or the quantized model is unavailable in their model zoo.

*Table 8.* C4 perplexity ($\downarrow$) of quantized models under 4-bit and 3-bit. GANQ outperforms state-of-the-art methods.

| Method | Bit-width | OPT | | | | | LLaMA | | |
|---|---|---|---|---|---|---|---|---|---|
| | | 125M | 350M | 1.3B | 2.7B | 6.7B | 7B | 2-7B | 3-8B |
| Full | 16 | 26.56 | 22.59 | 16.07 | 14.34 | 12.71 | 7.34 | 7.26 | 9.45 |
| RTN | 4 | 33.89 | 26.21 | 27.49 | 18.83 | 14.37 | 8.12 | 8.16 | 13.35 |
| GPTQ | 4 | 29.08 | 24.64 | 17.00 | 14.99 | 13.18 | 8.83 | 7.87 | 51.33 |
| OmniQuant | 4 | 28.76 | 23.85 | 16.85 | 14.93 | 13.10 | 7.66 | 7.77 | – |
| GANQ | 4 | **27.72** | **23.47** | **16.54** | **14.71** | **12.96** | **7.52** | **7.47** | **10.23** |
| RTN | 3 | 8.3e2 | 55.15 | 6.5e3 | 1.0e4 | 5.0e3 | 20.78 | 5.3e2 | 5.7e2 |
| GPTQ | 3 | 42.14 | 30.90 | 21.52 | 18.24 | 17.00 | 22.28 | 11.67 | 70.53 |
| OmniQuant | 3 | 36.37 | **28.82** | 19.61 | 19.10 | 15.51 | 8.75 | 9.38 | – |
| GANQ | 3 | **33.59** | 29.46 | **18.46** | **16.43** | **13.68** | **8.20** | **8.20** | **12.88** |

The results in Table 9 present the PTB perplexity of various quantized models under 4-bit and 3-bit configurations across different model sizes. As shown, GANQ outperforms baseline methods such as RTN, GPTQ, and OmniQuant across all configurations. For 4-bit quantization, GANQ achieves the lowest perplexity across OPT models with notable improvements. GANQ also demonstrates strong performance with 3-bit quantization, maintaining competitive perplexity reductions across model sizes. On OPT-6.7B, GANQ's perplexity is 16.91, compared to 21.63 for GPTQ and 21.52 for OmniQuant. These results highlight GANQ's effectiveness in both 4-bit and 3-bit quantization, achieving substantial perplexity reductions across a range of model sizes.

LLaMA-7B and LLaMA-2-7B perform similarly to the much smaller OPT-125M model in full-precision FP16 configuration on the PTB dataset. Specifically, the FP16 versions of LLaMA-7B (41.15) and LLaMA-2-7B (37.91) do not achieve significantly better perplexity than the OPT-125M model (38.99), highlighting the relative inefficiency of LLaMA models on this dataset. Therefore, we focus on reporting results for OPT models, which demonstrate stronger performance in this context.

*Table 9.* PTB perplexity (↓) of quantized models under 4-bit and 3-bit. GANQ outperforms state-of-the-art methods.

| Method | Bit-width | OPT | | | | |
|---|---|---|---|---|---|---|
| | | 125M | 350M | 1.3B | 2.7B | 6.7B |
| Full | 16 | 38.99 | 31.07 | 20.29 | 17.97 | 15.77 |
| RTN | 4 | 53.88 | 36.79 | 75.40 | 32.40 | 18.86 |
| GPTQ | 4 | 45.45 | 34.33 | 22.04 | 19.19 | 16.58 |
| OmniQuant | 4 | 42.53 | 33.80 | 21.79 | 19.00 | 16.18 |
| GANQ | 4 | **40.75** | **33.21** | **21.06** | **18.73** | **15.95** |
| RTN | 3 | 1.4e3 | 87.20 | 1.5e3 | 1.2e4 | 5.4e3 |
| GPTQ | 3 | 72.91 | 47.17 | 31.94 | 25.63 | 21.63 |
| OmniQuant | 3 | 59.56 | **42.65** | 26.87 | 29.82 | 21.52 |
| GANQ | 3 | **55.67** | 44.58 | **24.27** | **21.28** | **16.91** |

*Table 10.* Perplexity (↓) of quantized LLaMA-3.2 models on WikiText2 and C4. GANQ outperforms state-of-the-art methods.

| Dataset | Method | Bit-width | 1B | 3B | 1B-Instruct | 3B-Instruct |
|---|---|---|---|---|---|---|
| | FP16 | 16 | 9.76 | 7.81 | 13.16 | 11.05 |
| | RTN | 4 | 18.08 | 10.53 | 22.91 | 15.58 |
| | GPTQ | 4 | 24.07 | 6.0e3 | 18.40 | 4.5e3 |
| | OmniQuant | 4 | 12.90 | 8.87 | 16.31 | 12.33 |
| WikiText-2 | GANQ | 4 | **10.78** | **8.35** | **14.36** | **11.99** |
| | RTN | 3 | 2.6e3 | 4.8e2 | 7.0e3 | 5.7e2 |
| | GPTQ | 3 | 1.68e2 | 6.3e3 | 1.1e2 | 3.3e3 |
| | OmniQuant | 3 | 4.3e2 | 14.82 | 34.89 | 19.55 |
| | GANQ | 3 | **15.91** | **10.85** | **21.32** | **15.84** |
| | FP16 | 16 | 14.02 | 11.34 | 21.30 | 16.50 |
| | RTN | 4 | 27.98 | 15.89 | 35.23 | 22.15 |
| | GPTQ | 4 | 41.27 | 2.1e3 | 27.99 | 3.2e3 |
| | OmniQuant | 4 | 18.34 | 13.08 | 25.20 | 18.02 |
| C4 | GANQ | 4 | **15.53** | **12.21** | **22.81** | **17.35** |
| | RTN | 3 | 2.9e3 | 4.1e2 | 6.7e3 | 5.3e2 |
| | GPTQ | 3 | 1.5e2 | 1.6e3 | 1.2e2 | 2.0e3 |
| | OmniQuant | 3 | 3.9e2 | 19.96 | 41.43 | 24.80 |
| | GANQ | 3 | **22.09** | **15.36** | **29.18** | **21.30** |

Table 10 presents the perplexity of quantized LLaMA-3.2 models on WikiText-2 and C4 under 4-bit and 3-bit configurations. GANQ consistently outperforms baseline methods (RTN, GPTQ, OmniQuant) across all model variants and bit-widths. Our systematic optimization framework for layer-wise non-uniform quantization allows GANQ to adapt effectively to varying weight distributions, unlike baseline methods, which demonstrate greater sensitivity and instability, especially at 3-bit quantization.

