# OpenReview forum: "GANQ: GPU-Adaptive Non-Uniform Quantization for Large Language Models"
_ICML.cc/2025/Conference — ICML 2025 poster_

### Official Review · Reviewer_V6cW · 2025-03-13

**Overall Recommendation:** 3

**Summary:**

The paper proposes GANQ, a GPU-Adaptive Non-Uniform Quantization framework leveraging lookup table (LUT)-based mixed-precision GEMM for efficient deployment of Large Language Models (LLMs). GANQ introduces a post-training quantization optimization algorithm to effectively solve LUT-based quantization objectives, significantly reducing both quantization cost and quantization errors. Experiments show GANQ outperforms existing methods in reducing perplexity gaps at 3-bit and 4-bit quantization, achieving up to 2.57× inference speedup on an NVIDIA RTX 4090 GPU.

**Claims And Evidence:**

Please refer to "*Methods And Evaluation Criteria*".

**Essential References Not Discussed:**

N/A

**Experimental Designs Or Analyses:**

Please refer to "*Methods And Evaluation Criteria*".

**Methods And Evaluation Criteria:**

Strengths:
+ Introduces GANQ, an innovative GPU-adaptive, LUT-based non-uniform quantization method, demonstrating clear perplexity improvements for quantized OPT and Llama models.
+ Provides promising inference efficiency improvements (up to 2.57× speedup) on real GPU hardware (NVIDIA RTX 4090).

Weaknesses:
- Accuracy evaluations are limited, lacking tests on more recent models (e.g., Llama-3.1), long-context scenarios (LongBench), and reasoning-intensive benchmarks (e.g., GSM8K).
- Efficiency benchmarks only compare against GPTQ, with reported performance gains notably inconsistent (lower) compared to prior literature where GPTQ can gain 4x higher throughput.
- Experimental setups for latency measurement lack critical details such as batch size and input token length, hindering clarity, reproducibility, and proper interpretation of the results.

**Other Comments Or Suggestions:**

N/A

**Other Strengths And Weaknesses:**

N/A

**Questions For Authors:**

Please refer to "*Methods And Evaluation Criteria*".

**Relation To Broader Scientific Literature:**

This paper contributes to LLM quantization for efficient LLM inference.

**Theoretical Claims:**

Please refer to "*Methods And Evaluation Criteria*".

---

> ### Author Rebuttal · Authors · 2025-04-01
>
> Thanks for the feedback. Below, we address your concerns one by one.
>
> **Response to Weakness 1**
>
> We acknowledge the value of broader evaluation and have conducted additional experiments.
>
> Below are WikiText PPL results for **LLaMA-3.2** models, using the same settings as in the paper.
>
> ||1B|3B|1B-Instruct|3B-Instruct
> |-|:-:|:-:|:-:|:-:
> |FP16|9.76|7.81|13.16|11.05
> |(4-bit / 3-bit)
> |RTN|18.08 / 2.6e3|10.53 / 4.8e2|22.91 / 7.0e3|15.58 / 5.7e2
> |GPTQ|24.07 / 1.7e2|6.0e3 / 6.3e3|18.40 / 1.1e2|4.5e3 / 3.3e3
> |OmniQuant|12.90 / 4.3e2|8.87 / 14.82|16.31 / 34.89|12.33 / 19.55
> |GANQ|**10.78** / **15.91**|**8.35** / **10.85**|**14.36** / **21.32**|**11.99** / **15.84**
>
> These results confirm that GANQ consistently outperforms baselines with superior and more stable performance. While OmniQuant learns the quantization factors and is relatively robust, it still fails in some case (i.e, 3-bit for LLaMA-3.2-1B). RTN and GPTQ show significant sensitivity to weight distribution and degradation, with GPTQ notably unstable on 3B models. To explore this, we test GPTQ with a group size of 128 to mitigate outlier effects (same table head):
>
> ||||||
> |-|:-:|:-:|:-:|:-:
> |GPTQ (g128)|15.37 / 72.00|1.1e2 / 1.1e3|16.03 / 28.31|49.14 / 6.9e2
>
> Using a group size of 128 improves GPTQ's performance on 1B models but still performs poorly on 3B models, highlighting its sensitivity to weight distribution.
>
> We initially did not include long-context or reasoning-intensive tasks as we evaluated base models without task-specific fine-tuning (e.g., LLaMA-2-7B). Following standard practice [1,2], we used Table 3 tasks to assess general capabilities.
>
> We test LongBench and GSM8K on LLaMA-3.2 1B/3B-Instruct and their quantized versions.
>
> ||1B-Instruct||3B-Instruct||
> |-|-|-|-|-|
> ||LongBench|GSM8K (%)|LongBench|GSM8K (%)
> |FP16|11.5|32.90|12.7|64.97
> |RTN|0.2|4.17|12.1|37.68
> |GPTQ|error|11.14|error|~0.0
> |OmniQuant|8.9|16.53|11.5|54.74|
> |GANQ|11.7|27.75|12.5|60.50
>
> GANQ consistently outperforms baseline. For GPTQ, consistent with PPL results on 3B-Instruct, GSM8K accuracy is near zero. On LongBench, GPTQ failed with a "skip_special_tokens" error, despite identical settings across all methods and use of the official toolkit [5].
>
> We plan to incorporate these results into the revision.
>
> **Response to Weakness 2**
>
> In Table 5, we compare the efficiency of dequantization-based and LUT-based methods. Following [3], we use GPTQ as a representative baseline for dequantization-based method, as similar methods like RTN and OmniQuant can generally share the same inference kernel and offer comparable efficiency. We will clarify this in the revision.
>
> We acknowledge that the GPTQ efficiency reported in Table 5 is lower than that in the original GPTQ paper. This discrepancy is mainly due to differences in the inference kernel, hardware, and model size, with the kernel likely being the key factor.
>
> As noted in Section 4.1, we followed [1] and used the GPTQ-for-LLaMA [2] inference kernel for GPTQ-quantized models. The reported results reflect actual observations. However, since [2] is implemented in Triton, and our experiments were conducted on an NVIDIA RTX 4090, potential incompatibilities or limited hardware support may explain the slower inference in our setup.
>
> As for metric, as noted in Section 4.1, we followed [1,3], using a batch size of 1 to generate 1024-token sequences and reporting total CUDA time.
>
> To further validate this, we performed an additional test using GPTQ's official CUDA kernel implementation [1] (available only for 3-bit quantization and OPT models). The results (total CUDA time in seconds) are as follows:
>
> ||FP16|GPTQ(3bit)|GANQ(3bit)
> |-|-|-|-|
> |OPT-6.7B|16.784|6.543|6.519
>
> The results are consistent. We will incorporate these findings and clarifications into the revision to resolve potential confusion clearly.
>
> Finally, we would like to emphasize that our main contribution is the proposed LUT-based non-uniform quantization method, which improves quantization accuracy. For inference efficiency, it can similarly benefit from advances in LUT-based kernel implementations, as dequantization-based methods do with their kernels.
>
> **Response to Weakness 3**
>
> As detailed in Section 4.1 (Lines 313–318, right column) and reiterated in Response to Weakness 2, we follow the evaluation setup used in prior works [1,3] to measure inference latency. Specifically, we report the total CUDA time required for the model to generate 1024 tokens, using a batch size of 1 and no initial input tokens. We will make these settings more explicit in the revised paper to improve clarity and reproducibility.
>
> [1] Frantar, Elias, et al. "Gptq: Accurate post-training quantization for generative pre-trained transformers."
>
> [2] Sun, Mingjie, et al. "A simple and effective pruning approach for large language models."
>
> [3] Kim, Sehoon, et al. "Squeezellm: Dense-and-sparse quantization."
>
> [4] https://github.com/qwopqwop200/GPTQ-for-LLaMa
>
> [5] https://github.com/THUDM/LongBench

---

> > ### Comment · Reviewer_V6cW · 2025-04-05
> >
> > - Weakness 2: The authors' focus on the 3-bit latency measurement in their rebuttal misses a key point. The primary comparison for efficiency in Table 5 is based on 4-bit quantization, and I would expect to see evaluation results for that case instead. The authors' response seems to sidestep the central issue. In the original GPTQ implementation (or the Marlin library, TinyChat library, TensorRT-LLM), GPTQ (or any other 4-bit weight-only g128 quantization method e.g., AWQ) achieves nearly 4x higher throughput. In contrast, using the numbers from Table 5, GANQ only achieves about a 2x speedup, which is comparable to the speedup seen with 8-bit quantization. This highlights a significant discrepancy in efficiency that needs further explanation.
> >
> > - Weakness 3: In response to Reviewer QPHs, I am not requesting results based on larger batch sizes; rather, I am pointing out that the evaluation setups themselves are missing from the paper (as they only mention 1024 generation tokens). These details are crucial for understanding the context of the measurements. Additionally, both context length and generation length play a significant role in determining the computational characteristics of LLM inference, as they directly impact the proportion of attention operations, especially in long-context tasks and reasoning models that rely on chain-of-thoughts.

---

> > > ### Author Response · Authors · 2025-04-05
> > >
> > > Thank you for your thoughtful follow-up. Your insights, together with Reviewer QPHs', greatly help us clarify the key factors behind LLM inference speedup from model compression.
> > >
> > > **Response to Weakness 2**
> > >
> > > We attribute the speedup differences mainly to kernel implementation, hardware variations (e.g., FP16 cross-GPU overhead), the model, and the generation length, affecting memory- vs. compute-bound behavior (as noted by you and Reviewer QPHs). Table 5 accurately reflects performance in our setup, with **kernel choice** being the primary factor for the 4-bit speedup gap. Thus, we previously verified this using an alternative GPTQ kernel (only 3-bit support). Below, we would like to make it more clear and finally replicate AWQ's 4-bit kernel performance in our setting.
> > >
> > > **1: Kernel Implementation Impact**
> > >
> > > Following [1], we initially used [2] as the GPTQ kernel, which showed a lower speedup due to suboptimal GPU efficiency in our experimental environment.
> > >
> > > To validate this, we performed experiments using GPTQ’s official CUDA kernel (currently only for OPT models in 3-bit). Here are results (batch size=1, seq_len=1024):
> > >
> > > ||FP16|GPTQ-3bit|GANQ-3bit
> > > |-|-|-|-|
> > > OPT-6.7B|16.784|6.543|6.519
> > >
> > > GPTQ shows about 2.5$\times$ speedup, which is much improved compared with [2] but still lower than the 3.25$\times$ or 4.53$\times$ reported in GPTQ paper. We attribute the remaining gap to differences in hardware (e.g., number of GPUs), model size, and generation length, as detailed in the next section.
> > >
> > > **2. Hardware, Model Size, Generation Length Impact**
> > >
> > > The GPTQ paper's higher speedups (3.25$\times$ or 4.53$\times$) were obtained under conditions vastly different from ours:
> > >
> > > |Bit|Speedup|Model|Length|GPU for FP16|GPU for 3-bit|
> > > |-|-|-|-|-|-|
> > > 3|3.25|OPT-175B|128|8 A6000|2 A6000
> > > 3|4.53|OPT-175B|128|5 A100| 1 A100
> > >
> > > The critical difference here is that their FP16 baseline required multiple GPUs (incurring significant cross-GPU communication overhead), making their setup inherently memory-bound and thus more advantageous to quantization. Our experiments were performed entirely on a single RTX 4090, with smaller model size (OPT-6.7B), longer generation lengths (1024), and no cross-GPU overhead. This naturally reduces the relative benefit from quantization.
> > >
> > > We explicitly demonstrate this impact through additional benchmarking for OPT-6.7B across different generation lengths (single RTX 4090, batch size=1, CUDA time in seconds):
> > >
> > > |Length|FP16|GPTQ(3-bit)|GANQ(3-bit)
> > > |:-:|-|-|-|
> > > |64|0.986|0.353|0.329
> > > |256|4.000|1.464|1.383
> > > |1024|16.790|6.553|6.528
> > >
> > > These results clearly show that longer generation sequences reduce the observed speedup (e.g., from ~2.79 at length 64 down to ~2.56 at length 1024 for GPTQ), further explaining why our reported speedup for GPTQ in 3-bit is lower.
> > >
> > > **3. AWQ Kernel Validation**
> > >
> > > Following your suggestion, we use AWQ’s kernel, which reports a 2.79–3.3$\times$ speedup on a single RTX 4090 with batch size 1 and sequence length 200 for various models in the AWQ paper. Below are our results on LLaMA-7B (batch size = 1, sequence length = 200):
> > >
> > > ||CUDA Time (s)|Speedup ($\uparrow$)|Peak Memory (GB, $\downarrow$)
> > > |-|-|-|-|
> > > FP16|3.26|1.00|12.66
> > > AWQ|1.18|2.76|3.87
> > > GANQ|1.50|2.17|3.76
> > >
> > > AWQ’s measured speedup (~2.76$\times$) aligns closely with the original AWQ paper’s results for 7B models.
> > >
> > > Again, this highlight the pivotal role of **kernel optimization** for quantization efficiency, further clarifying the discrepancy. We observed gains moving from GPTQ-for-LLaMA [2] to the GPTQ CUDA kernel, and further to AWQ’s CUDA kernel, despite similar quantization schemes. The kernel from [1] we used performs well but may still leaves room for improvement.
> > >
> > > We emphasize that our primary contribution is in improving LUT-based quantization accuracy rather than kernel-level optimization. Nonetheless, our method remains compatible with future LUT-based kernel improvements (e.g., 2–4$\times$ in [3], up to 6.93$\times$ in [4]) can directly amplify GANQ’s performance. **We believe our work, alongside advances in kernel engineering, can jointly drive meaningful progress in the field.**
> > >
> > > **Response to Weakness 3**
> > >
> > > We fully agree on the importance of clearly stating experimental setups, including hardware, inference kernels, batch size, and generation lengths. We'll explicitly clarify these details as well as above analysis in our revision.
> > >
> > > Finally, we plan to revise Table 5 in the paper to explicitly highlight GANQ’s compatibility with existing LUT-based kernels and emphasize that our focus remains on more accurate quantization rather than kernel optimization.
> > >
> > > Thank you again for your time and thoughtful insights!
> > >
> > > [1] Kim, Sehoon, et al. "Squeezellm: Dense-and-sparse quantization."
> > >
> > > [2] https://github.com/qwopqwop200/GPTQ-for-LLaMa
> > >
> > > [3] Guo, Han, et al. "Fast matrix multiplications for lookup table-quantized llms."
> > >
> > > [4] Mo, Zhiwen, et al. "Lut tensor core: Lookup table enables efficient low-bit llm inference acceleration."

---

### Official Review · Reviewer_QPHs · 2025-03-15

**Overall Recommendation:** 5

**Summary:**

This paper proposes a look-up-table based non-uniform quantization algorithm for LLMs. The algorithm is based on mixed-integer quadratic programming, with a mathematical proof that the authors deduce. The results show that the proposed method has better accuracy than state of the art methods on 4-bits and 3-bits quantization. The authors also provide a CUDA implementation that shows more than 2x speedup over fp16 baseline on NVIDIA RTX 4090 GPU.

**Claims And Evidence:**

- The claims of obtaining higher accuracy compared to other methods have been supported by strong results

**Essential References Not Discussed:**

N/A

**Experimental Designs Or Analyses:**

- I believe accuracy comparisons are thorough and done properly
   - However, I am concerned about the context length of perplexity, as different papers have evaluated perplexity with different context length. So need to double check that context length is consistent in the comparisons.
- Though the CUDA implementation of the proposed method has decent speedup, I am concerned that the measured GPTQ CUDA speedup was around 0.3x (i.e., 3 times slower than the baseline), while Table 6 of the GPTQ paper shows speedups of up to 4.53x, albeit on a different GPU, and different LLM architecture. If there is an issue in reproducing similar speedups claimed by the GPTQ paper, I suggest not to include those measurements in the table

**Methods And Evaluation Criteria:**

- Paper evaluated different model families (OPT and Llama), diferent generations (LLama1, Llama2, and Llama3), and different sizes (8B, 70B, etc.)
   - I would suggest adding:
        - other more recent families (e.g., Mistral or DeepSeek)
        - other sizes of recent models (e.g., Llama 3.2 1B, Llama3 70B). Though I understand that the authors may not be able to fit large models if they only have access to a RTX 4090 GPU
- Compared with a good number of approaches that use or don't use outlier mitigation techniques

**Other Comments Or Suggestions:**

- Equation 5: Please provide citation to a paper or textbook that explains how the closed form solution could be obtained
- Line 286, Column 2: "by reporting perplexity on language generation tasks": my understanding generation tasks don't measure perplexity. I would prefer to re-word it to "perlexity on language datasets"
    - Please specify context length of perplexity evaluation

**Other Strengths And Weaknesses:**

Strengths:
- Usage of mixed integer quadratic programming is probably a novelty in the quantization field. Though, I might be wrong.
- The proposed algorithm is based on rigorous mathematical proof
- The proposed algorithm was optimized for parallelized compute platforms and implemented on GPU
- "GANQ consistently outperforms baseline methods such as RTN, GPTQ, and OminiQuant across all configurations"
- The proposed algorithm doesn't require a lot of memory, unlike approaches like SqueezeLLM and OmniQuant, and hence can quantize larger models on smaller GPUs.

**Questions For Authors:**

- Is the algorithm similar to GPTQ's algorithm, in the sense that both decide to quantize each element in a row sequentially?

**Relation To Broader Scientific Literature:**

The paper compared with latest state of the art quantization algorithms (GPTQ, AWQ, OmniQuant, SqueezeLLM)

**Theoretical Claims:**

- I don't claim that I have fully understood the mathematical proof.

---

> ### Author Rebuttal · Authors · 2025-03-31
>
> Thank you for your thoughtful feedback and strong support. We appreciate your recognition of our contributions and provide detailed responses below.
>
> **Comment 1:** *The context length of perplexity.*
>
> Thanks for pointing this out. We use a sequence length of 2048 for all models and methods, following prior work, and will clarify this in the revision.
>
> **Suggestion 1:** *More recent models and model families/sizes:*
>
> Thank you for the suggestion. To further validate GANQ, we conducted additional experiments on the latest LLaMA-3.2 models. Below are the WikiText PPL results (sequence length 2048, same setting as in our paper).
>
> ||1B|3B|1B-Instruct|3B-Instruct
> |-|:-:|:-:|:-:|:-:
> |FP16|9.76|7.81|13.16|11.05
> |(4-bit / 3-bit)
> |RTN|18.08 / 2.6e3|10.53 / 4.8e2|22.91 / 7.0e3|15.58 / 5.7e2
> |GPTQ|24.07 / 1.7e2|6.0e3 / 6.3e3|18.40 / 1.1e2|4.5e3 / 3.3e3
> |OmniQuant|12.90 / 4.3e2|8.87 / 14.82|16.31 / 34.89|12.33 / 19.55
> |GANQ|**10.78** / **15.91**|**8.35** / **10.85**|**14.36** / **21.32**|**11.99** / **15.84**
>
> The additional results confirm that GANQ consistently achieves superior and more stable quantization performance. Our systematic optimization framework for layer-wise non-uniform quantization allows GANQ to adapt effectively to varying weight distributions, unlike baseline methods, which demonstrate greater sensitivity and instability, especially at 3-bit. We will add these findings in the revision.
>
> Regarding the suggestion to evaluate more model families (e.g., DeepSeek) and larger scales (e.g., LLaMA-3 70B), we appreciate your understanding that such experiments are limited by our hardware (RTX 4090 GPU). That says, since GANQ operates row-wise on linear layers and is designed to adapt to diverse weight distributions, we expect it to generalize well to other architectures and larger models. We plan to extend our evaluations to broader model families and larger scales in the future work.
>
> **Comment 2:** *GPTQ CUDA speedup measurements*
>
> We acknowledge that the GPTQ efficiency reported in Table 5 is lower than that in the original GPTQ paper. This discrepancy is mainly due to differences in the inference kernel, hardware, and model size, with the kernel likely being the key factor.
>
> As noted in Section 4.1, we followed prior work [1] and used the GPTQ-for-LLaMA [2] inference kernel for GPTQ-quantized models. The reported results reflect actual observations. However, since [2] is implemented in Triton, and our experiments were conducted on an NVIDIA RTX 4090, potential incompatibilities or limited hardware support may explain the slower inference in our setup.
>
> To further validate this, we performed an additional test using GPTQ's official CUDA kernel implementation [3] (available only for 3-bit quantization and OPT models). The results (total CUDA time in seconds) are as follows:
>
> ||FP16|GPTQ (3bit)|GANQ (3bit)
> |-|-|-|-|
> |OPT-6.7B|16.784|6.543|6.519
>
> The results are consistent. We will incorporate these findings and clarifications into the revision to resolve potential confusion clearly.
>
> **Suggestion 2:** *The derivation of the closed-form solution of Equation 5*
>
> We will add the detailed derivation of Equation 5 in the appendix of the revision.
>
> Let
> $$
> f(\\mathbf{T}_i)=\\|\\mathbf{W}_i\\mathbf{X} \\mathbf{T}_i\\mathbf{S}_i^{k+1}\\mathbf{X}\\|^2.
> $$
> by the first-order optimality, let
> $$
> \\nabla f(\\mathbf{T}_i)=2(\mathbf{W}_i\\mathbf{X}- \\mathbf{T}_i\\mathbf{S}_i^{k+1}\\mathbf{X})\\mathbf{X}^\\top(\\mathbf{S}_i^{k+1})^\\top=0,
> $$
> we have
> $$
> \\mathbf{T}_i^{k+1}=\\mathbf{W}_i \\mathbf{XX}^\\top (\\mathbf{S}_i^{k+1})^\\top ((\\mathbf{S}_i)^{k+1}\\mathbf{XX}^\\top (\\mathbf{S}_i^{k+1})^\\top)^\\dagger,
> $$
> where $(\\cdot)^\\dagger$ denotes the Moore-Penrose inverse.
>
> **Corrections**
>
> We agree that "perplexity on language datasets" is a clearer phrase and will adopt this wording in the revision.
>
> **Response to Question for Authors**
>
> Although both GPTQ and our proposed method (GANQ) quantize model weights in a row-wise manner, the strategies differ significantly. GPTQ uses a greedy, sequential element-wise quantization based on the Optimal Brain Surgeon framework [4], quantizing one element at a time and updating subsequent weights to minimize reconstruction error.
>
> In contrast, our method models LUT-based non-uniform quantization as a mixed-integer quadratic program. Rather than sequentially quantizing elements, we employ an alternating optimization for an entire row with a closed-form solution for the T-subproblem and an efficient back-substitution algorithm for the S-subproblem, enabling more systematic and effective quantization than GPTQ.
>
>
> [1] Kim, Sehoon, et al. "Squeezellm: Dense-and-sparse quantization."
>
> [2] https://github.com/qwopqwop200/GPTQ-for-LLaMa
>
> [3] Frantar, Elias, et al. "Gptq: Accurate post-training quantization for generative pre-trained transformers."
>
> [4] Hassibi, Babak, and David Stork. "Second order derivatives for network pruning: Optimal brain surgeon."

---

> > ### Comment · Reviewer_QPHs · 2025-04-04
> >
> > I would like to thank the authors for their detailed rebuttal.
> > I have read all the reviews and their corresponding rebuttals and I would like to keep my score.
> > I believe authors have comprehensively responded to the requests by the reviewers.
> >
> > Regarding the request by Reviewer V6cW to include speedups on larger batch sizes and longer sequence lengths: my understanding is that weight-only quantization (that is the scope of this paper) optimizes memory bound processes, which is the case of LLM autoregressive decoding for batch size 1 and moderate sequence length. For larger batch sizes or very large sequence lengths, autoregressive decoding becomes more of a compute bound process and weight-only quantization may not speed it up (in some cases it may even slow it down). Hence, most papers on weight-only quantization don't evaluate speedups for batch size > 1. (On the other hand, approaches on weight+activation quantization can speed up such compute bound processes).

---

> > > ### Author Response · Authors · 2025-04-04
> > >
> > > Thank you for your kind words and for taking the time to carefully read our rebuttal and the reviewers’ responses. We greatly appreciate your feedback and constructive comments.
> > >
> > > Your clarification regarding the distinction between weight-only and weight+activation quantization is very helpful. We will highlight this distinction clearly in the revision. Besides, we believe that how to extend our method to "weight+activation quantization" is a valuable point for future research.
> > >
> > > Thank you once again for your support!

---

### Official Review · Reviewer_xu7k · 2025-03-18

**Overall Recommendation:** 3

**Summary:**

The paper present GANQ, a GPU-Adaptive Non-Uniform Quantization technique specifically tailored for efficient inference of Large Language Models (LLMs). GANQ introduces a principled optimization model based on Mixed-Integer Quadratic Programming (MIQP) to achieve layer-wise quantization using a Lookup Table (LUT) approach. It leverages GPU acceleration to efficiently handle the computational complexity. Extensive evaluations demonstrate GANQ’s significant reduction in perplexity relative to state-of-the-art methods across various models and datasets, achieving a notable inference speedup (up to 2.57X on NVIDIA RTX 4090)

**Claims And Evidence:**

yes

**Essential References Not Discussed:**

no

**Experimental Designs Or Analyses:**

yes

**Methods And Evaluation Criteria:**

yes

**Other Comments Or Suggestions:**

## update after rebuttal: I appreciate the responses from the author. I am happy to increase my score.

**Other Strengths And Weaknesses:**

strengths
1. GANQ clearly formulates non-uniform quantization as an MIQP problem, allowing systematic optimization rather than heuristic approaches.
2. Demonstrates superior perplexity improvements across both 3-bit and 4-bit quantization, and outperforming established quantization methods.
3. exploits GPU parallelism by decomposing the original optimization into highly parallelizible subproblems.

weakness:
1. The method relies heavily on Cholesky decomposition, potentially limiting its applicability if XX^T matrices are ill-conditioned.
2. Limited details on how the MIQP is solved on GPU, there are mature toolkits for the optimization problems such as Gurobi or CPLEX, more discussions are needed.
3. Table 2 does not include performance of AWQ and suqeezeLLM, better to have a more comprehensive experiements.

**Questions For Authors:**

How sensitive is GANQ’s quantization accuracy and optimization efficiency to the conditioning of the Cholesky decomposition matrix XX^T? Have you explored alternative numerical methods for stability?

**Relation To Broader Scientific Literature:**

Novel approach to use MIQP to find the best layer wise quantization strategy

**Theoretical Claims:**

yes

---

> ### Author Rebuttal · Authors · 2025-03-28
>
> Thank you for the insightful comments and recognition of GANQ's contributions. Below, we address the concerns raised.
>
> ## Concern 1: Reliability of Cholesky Decomposition and Sensitivity to Preconditioning of $XX^\top$
>
> **Response to Weakness 1**
>
> Yes, our method relies on Cholesky decomposition to efficiently solve the S-subproblem. When $XX^\top$ is ill-conditioned, we apply preconditioning to ensure the decomposition remains feasible, maintaining the effectiveness of our framework. This is a standard technique in numerical linear algebra.
>
> **Response to Questions for authors**
>
> Yes, we implement an adaptive preconditioning method for $XX^\top$ and examine its impact on quantization accuracy and efficiency below.
>
> Remark 3.1 introduces preconditioning $XX^\top$ via $XX^\top+\lambda I$, with $\lambda>0$. In practice, we adopt an adaptive method that enforces diagonal dominance, ensuring positive definiteness without manual tuning $\lambda$.
>
> A symmetric matrix $A$ is positive definite if it is diagonally dominant with positive diagonal entries, i.e., $|a_{ii}| \geq \sum_{j \neq i} |a_{ij}|$ for all $i$. Let $\Sigma = XX^\top$, with $\Sigma_{ii} \geq 0$. For each row $i$, we compute the offset vector $\delta$ as follows:
> $$\delta_i = \max\left(\sum_{j=1}^n|\Sigma_{i,j}|-2\Sigma_{i,i},\ 10^{-8}\right),
> $$where $\Sigma_{i,j}$ denotes the element at the $ i$-th row and $ j$-th column of $ \Sigma $. Then, perform
> $$L=\mathrm{Cholesky}(\Sigma+\mathrm{Diag}(\delta)),
> $$where $\mathrm{Diag}(\delta)$ constructs a diagonal matrix from the vector $\delta$, and $L$ is the lower-triangular Cholesky factor.
>
> To assess sensitivity to preconditioning (fixed $\lambda$ and adaptive) before Cholesky decomposition, we run 4-bit quantization experiments on OPT-125M and report PPL on WikiText.
> ||$\lambda=0.5$|$\lambda=1.0$|$\lambda=10.0$|$\lambda=40.0$|$\lambda=100$|Diagonally dominant
> |-|-|-|-|-|-|-|
> |PPL|29.14|29.04|28.98|29.05|29.09|28.58
>
> For a clear comparison on quantization accuracy, we also show the baseline methods again below.
> ||Full (FP16)|RTN|GPTQ|OmniQuant
> |-|-|-|-|-|
> |PPL|27.66|37.11|31.08|30.98
>
> The results show that quantization accuracy is largely **insensitive** to the choice of preconditioning. The adaptive method achieve the best PPL (28.58), with fixed $\lambda$ yielding similar results. All results outperforms baselines, confirming the robustness.
>
> In terms of efficiency, our preconditioning uses simple operations (e.g., summation, diagonal adjustments) with no noticeable impact on overall efficiency.
>
> **We initially considered this an implementation detail or engineering trick and plan to include the discussion in the Appendix of the revised paper.**
>
> ## Concern 2: Details on GPU-Based MIQP Solver Implementation
>
> **Response to Weakness 2**
>
> - **Details on how the MIQP is solved on GPU**
>
> Section 3.2 details how we solve the MIQP model using an alternating direction framework and leverage GPU for efficient computation.
>
> As shown in Equation (5), the T-subproblem has a closed-form solution with matrix-vector multiplications, which are highly efficient on GPUs.
>
> As shown in Equations (6)–(20), Figure 2, and the corresponding text (Lines 272–274, right column; Lines 295–300, left column), we illustrate how the S-subproblem is solved. Exploiting row-wise independence, we stack $W_i$ and $T_i$ vectors and organize $S_i$ matrices into a tensor, enabling parallel computation across rows on GPUs.
>
> In summary, we group independent vectors and matrices into larger matrices and tensors, enabling operations like multiplication and summation to be efficiently accelerated on GPUs.
>
> - **The adoption of Gurobi or CPLEX**
>
> While Gurobi and CPLEX are powerful commercial solvers, they are designed on generic purpose for general optimization models, including MIQP. In contrast, our framework exploits the specific structure of the model under discussion, yielding greater efficiency than generic solvers that overlook such problem-specific properties.
>
> We attempted to use Gurobi to solve Equation (2) during rebuttal, a single-row subproblem with OPT-125M dimensions, but it produced no output within three minutes. In contrast, our GPU-accelerated framework quantizes the entire model in the same time.
>
> ## Concern 3: Comparison with AWQ and SqueezeLLM
>
> **Response to Weakness 3**
>
> We compared our method with AWQ and SqueezeLLM in Table 4, as both incorporate mechanisms for handling outliers. AWQ utilizes a quantization block size of 128 (a default setting in the AWQ paper), which helps mitigate the impact of outliers. A key contribution of SqueezeLLM is isolating outliers into a separate sparse matrix while retaining 10 FP16 rows. Accordingly, we categorize these methods as outlier-aware quantization approaches and include their results in Table 4 for a fair comparison. As shown in Table 4, our method, equipped with a similar outlier-handling mechanism, achieves better performance compared to both.

---

### Official Review · Reviewer_nv8K · 2025-03-20

**Overall Recommendation:** 4

**Summary:**

The paper proposes GANQ, a post-training non-uniform quantization method optimized for hardware-efficient mpGEMM for LLMs. GANQ is LUT-based weight-only quantization capable of handling outliers. The experimental results demonstrate that the proposed GANQ outperforms baselines and achieves up to 2.57 times speedup.

**Claims And Evidence:**

1. The paper proposes GANQ (GPU-Adaptive Non-Uniform Quantization), a post-training non-uniform quantization method.

2. GANQ uses GPU-adaptive optimization, improving efficiency and achieving several times speedup compared to baselines.

3. GANQ is a non-uniform quantization method capable of handling outliers, improving performance.

4. Experimental results confirm GANQ’s efficiency and performance.

**Essential References Not Discussed:**

All essential references are discussed.

**Experimental Designs Or Analyses:**

The experimental designs and analyses are clear and comprehensive. However, some suggestions could improve the paper.

1. The settings of GANQ\* (with outlier handling) are not explained, such as the ratio of outliers and quantized weights.

2. Some baselines are missing from the profiling comparison (Table 5) without explanation.

3. GANQ\* should be compared in profiling and quantization cost.

4. The ratio of outliers and quantized weights could be discussed in more detail, as different ratios affect performance and cost.

**Methods And Evaluation Criteria:**

The proposed method is evaluated on perplexity, CUDA time, peak memory, and quantization cost, providing a reasonable and comprehensive assessment of GANQ.

**Other Comments Or Suggestions:**

Line 134 (right column): a repetition of the word “measures“

**Other Strengths And Weaknesses:**

All strengths and weaknesses have been mentioned above.

**Questions For Authors:**

In Table 5 (CUDA time), the full model only performs matrix multiplication, while GANQ involves table lookup, weight replacement, and then matrix multiplication. What explains GANQ being faster than the full model?

**Relation To Broader Scientific Literature:**

The paper proposes a post-training non-uniform quantization method that improves efficiency and performance compared to previous works. The experimental results align with prior findings, showing that quantization can degrade perplexity while reducing memory demands.

**Theoretical Claims:**

The theoretical proof of GANQ is well-constructed, clear and easy to follow.

1. $m, n, p$ in line 147 and 148 (right column) are not introduced.

---

> ### Author Rebuttal · Authors · 2025-03-31
>
> Thanks for the review and helpful suggestions. We appreciate your positive feedback and address your concerns below.
>
> **Comment:** *$m,n,p$ in line 147 and 148 are not introduced.*
>
> **Response:** $m,n,p$ denote the dimensions in a linear layer. In LLMs, $m$ is the output dimension, $n$ the input dimension, and $p$ the total number of tokens processed (batch size $\times$ sequence length). These values vary by models and layers. For example, in LLaMA-7B:
> ||m|n
> |-|-|-|
> |$W_{k,q,v,o}$|4096|4096
> |$W_{gate,up}$|11008|4096
> |$W_{down}$|4096|11008
>
> We use 128 samples with a sequence length of 2048, so $p=128\times 2048$. This highlights the large scale of the problem and the efficiency of our method in solving it.
>
> **Suggestion 1:** *The settings of GANQ$^\*$ (with outlier handling) are not explained ...*
>
> **Response:** We describe GANQ$^\*$'s outlier-handling in Section 3.3 and detail it in Algorithm 2 (Appendix A), which separates outliers based on a ratio $r$, with the rest for quantization. As noted in Lines 365-367 (right), we typically set $r=0.5\\%$ for a fair comparison. We will further clarify this in the revision.
>
> **Suggestion 2:** *Some baselines are missing from the profiling comparison ...*
>
> **Response:** In Table 5, we compare the efficiency of dequantization-based and LUT-based methods. Following [1], we use GPTQ as a representative baseline for dequantization-based method, as similar methods like RTN and OmniQuant can generally share the same inference kernel and offer comparable efficiency. We will clarify this in the revision.
>
> Besides, as in [1], we use GPTQ-for-LLaMa [2] as the inference kernel for GPTQ, which currently supports 4-bit acceleration only. Thus, Table 5 includes only 4-bit GPTQ results. While briefly mentioned in Section 4.3, we will clarify it further in the revision and plan to add results with GPTQ kernels that support 3-bit acceleration.
>
> **Suggestion 3:** *GANQ$^\*$ should be compared in profiling ...*
>
> **Response:** We show the results for GANQ$^\*$ below:
>
> ||GANQ$^\*$|CUDA Time (s)|Speedup|Peak Memory (GB)
> |-|-|-|-|-|
> |OPT-6.7B|4-bit|10.39|1.61|5.13
> ||3-bit|10.73|1.56|4.39
> |LLaMA-7B|4-bit|9.82|1.82|4.16
> ||3-bit|8.85|2.02|3.32
>
> While GANQ$^\*$ offers better quantization quality via outlier handling, it incurs slightly higher inference time due to separate sparse matrix operations. Thus, the choice between GANQ and GANQ$^\*$ should depend on the desired trade-off between quality and efficiency.
>
> We note that 3-bit quantization for OPT-6.7B in GANQ$^\*$ leads to longer inference times than 4-bit, likely due to: (1) memory bandwidth reduction being the main speedup factor (as in our response to *Questions for Authors* below); (2) sparse outlier operations becoming a bottleneck in this case; and (3) 3-bit values misaligning with byte boundaries (INT8), causing entries to span bytes and adding indexing overhead.
>
> We will clearly present and discuss these findings in the revision.
>
> **Suggestion 4:** *The ratio of outliers and quantized weights could be discussed ...*
>
> **Response:** As in Tables 2 and 3, GANQ is already effective without outlier handling, outperforming baselines and close to FP16 models.
>
> As noted in Section 3.3, adding outlier handling can further improve quantization quality but increases inference time and memory, highlighting a trade-off. To illustrate this, we compare both the PPL and efficiency metrics in one table. For example, on LLaMA-7B:
>
> ||PPL on WikiText|CUDA Time (s)|Speedup|Peak Memory (GB)
> |-|-|-|-|-|
> |FP16|5.68|17.86|1.00|13.06
> |4-bit|5.83|8.46|2.11|4.14
> |4-bit+0.5%|5.76|9.82|1.82|4.16
>
> These results demonstrate a clear trade-off between quantization quality and efficiency. We will include this discussion in the appendix to highlight this balance.
>
> **Response to Questions For Authors**
>
> GANQ's speedup mainly arises from reduced memory bandwidth usage, which align with observation in other model compression methods (Figure 3 in [3]). By storing weights as 4-bit or 3-bit indices, GANQ greatly lowers memory traffic between global memory and CUDA cores compared to FP16.
>
> While GANQ introduces some overhead from table lookups and weight reconstruction, these are minimal. The lookup table are small (e.g., 16 entries, 32 bytes per row for 4-bit quantization), and thus reside in fast constant or shared memory, adding negligible latency. Weight reconstruction is efficiently fused with matrix multiplication in CUDA kernels, avoiding extra global memory accesses.
>
> In summary, GANQ’s modest overhead is far outweighed by memory savings, resulting in faster overall inference, especially on bandwidth-constrained GPUs.
>
> **Response to Other Suggestions**
>
> Thanks for catching the typo, we will fix it in the revision.
>
> [1] Kim, Sehoon, et al. "Squeezellm: Dense-and-sparse quantization."
>
> [2] https://github.com/qwopqwop200/GPTQ-for-LLaMa
>
> [3] Lin, Ji, et al. "Awq: Activation-aware weight quantization for on-device llm compression and acceleration."

---

### Official Review · Reviewer_WtKA · 2025-03-22

**Overall Recommendation:** 2

**Summary:**

This paper proposes a GPU-adaptive non-uniform quantization framework for LLMs. By formulating quantization as a mixed-integer optimization problem, the authors aim to achieve efficient low-bit inference on hardware that lacks native mixed-precision matrix multiplication support. Their approach relies on lookup tables (LUTs) rather than repeated dequantization. They introduce an iterative algorithm to solve for the best codebooks per layer in a way that (i) addresses outliers, and (ii) facilitates rapid table lookups. The paper reports strong perplexity results on several LLM architectures (OPT, LLaMA, etc.), along with speedups when compared to FP16 baselines.

## update after rebuttal

Thanks to the authors for the explanations provided in response to my questions, which largely address my initial concerns. However, the paper still requires significant revisions to offer a more comprehensive discussion and comparison with other LUT-based methods. It also needs more rigorous experiments, concrete evidence of real-world efficiency, and details on efficient implementation.

**Claims And Evidence:**

The main claims are that (a) non-uniform LUT-based quantization, solved through a GPU-friendly mixed-integer optimization procedure, can significantly improve perplexity at lower bitwidths versus uniform quantization baselines; (b) the proposed method is broadly compatible with outlier-handling strategies; and (c) it yields better throughput and memory savings.

These are overall supported by experiment results.

**Essential References Not Discussed:**

- Fast Matrix Multiplications for Lookup Table-Quantized LLMs [1]: The authors should discuss how their proposed LUT-based approach differs from or improves upon the matrix multiplication kernels and quantization schemes in that work.
- LUT Tensor Core: Lookup Table Enables Efficient Low-Bit LLM Inference Acceleration [2]: Though this work is cited, there needs a in depth discussion since this work also reports accuracy vs. speed trade-offs with LUT-based quantization. Drawing explicit comparisons would be valuable, especially for verifying whether the GPU-adaptive optimization here significantly outperforms simpler LUT-coded designs.

[1] https://arxiv.org/abs/2407.10960

[2] https://arxiv.org/abs/2408.06003

**Experimental Designs Or Analyses:**

The experimental design is approporate.

While there lacks real end-to-end performance numbers (throughput/latency in tokens/second or ms/token) directly compared to baseline quantization methods. Currently, memory usage numbers are mentioned selectively; a more explicit breakdown across methods and model sizes would clarify the trade-off.

**Methods And Evaluation Criteria:**

The core method is well-motivated and the criteria and benchmarks are widely used.

**Other Comments Or Suggestions:**

In tables, OminiQuant -> OmniQuant

**Other Strengths And Weaknesses:**

I summarize all strengths and weaknesses here:

Strengths:
- Well-motivated approach in bridging the hardware-software gap for efficient low-bit inference.
- The per-layer, parallelizable formulation is elegant and presumably implementable with moderate engineering effort.
- Strong empirical results on perplexity across multiple models.

Weaknesses:
- Limited demonstration of real-time inference improvements. In practice, adopters would want to see the latency/throughput gains across model sizes, batch sizes, and sequence lengths.
- Comparisons with more LUT-specific or hardware-accelerated methods are not deeply explored.
- Code availability is not explicitly offered. Releasing the code would help ensure reproducibility and allow the community to adopt the method more easily.

**Questions For Authors:**

See weaknesses.

**Relation To Broader Scientific Literature:**

This paper focus on LLM quantization, which is related to works like GPTQ, OmniQuant, AWQ, etc.

**Theoretical Claims:**

The primary theoretical discussion is the layer-wise formulation as a mixed-integer quadratic program and its subsequent decomposition. There does not appear to be any new theorems with formal proofs requiring correctness checks.

---

> ### Author Rebuttal · Authors · 2025-03-31
>
> Thank you for the thorough and insightful feedback. We would like to address your concerns point by point below.
>
> ## Clarification of References ([1, 2]) and Response to Weakness 2
> Thank you for highlighting the relevance of references "Fast Matrix Multiplications for Lookup Table-Quantized LLMs [1]" and "LUT Tensor Core: Lookup Table Enables Efficient Low-Bit LLM Inference Acceleration [2]". As their titles suggest, these works primarily contribute optimized LUT-based matrix multiplication kernels and tensor cores. In contrast, our primary contribution is a novel non-uniform quantization scheme tailored for LUT-based inference. Our method produces quantized representations that can leverage and complement the kernels or tensor cores proposed in [1,2]. Indeed, enabling such compatibility was part of our motivation, which we have explicitly mentioned in the Introduction (Lines 93-95, left column), where [2] has been cited.
>
> Due to different objectives, [1] and [2] focus on end-to-end inference speedup using custom kernels, while our work emphasizes quantization accuracy and compatibility with LUT-based kernels.
>
> For inference speed, we currently use the kernel from [3], as noted in Section 4.1. In the future, we plan to adopt newer LUT-based inference kernels like [1,2] to take advantage of their advancements.
>
> For quantization accuracy, to the best of our knowledge, [2] does not propose a new quantization scheme. Instead, it validates the LUT tensor core by precomputing lookup tables with operator fusion. For [1], it utilizes a learned NormalFloat quantization scheme. As noted in their Section 4.2: "*Based on our earlier experiments, we selected a group size of 64, which strikes a good balance between quality and speed.*" This means that every 64 elements share a lookup table. While this improves quantization quality, it increases memory overhead. For example, for LLaMA-3-8B's $W_{q}$ ($4096\times 4096$), a group size of 64 results in 64 ($4096/64=64$) lookup tables per row (each with 16 FP16 values for 4-bit quantization). In contrast, our method is initially designed to support one lookup table per row (i.e., channel-wise), which is much more memory-efficient. Below is the theoretical model compression ratios (0.5 byte for INT4, 2 byte for FP16):
> |[1]|Ours
> |-|-|
> |$\frac{0.5\times 4096^2+2\times 64\times 16 \times 4096}{2 \times 4096^2}=50\\%$|$\frac{0.5\times 4096^2+2\times 1\times 16 \times 4096}{2 \times 4096^2}=25.39\\%$
>
> Besides, as noted in Section 3.3, our method supports outlier-handling techniques by extracting outliers into a sparse matrix, balancing compression and quality. For LLaMA-3-8B, our method already outperforms [1] in 3-bit quantization with just 0.5% outlier separation. In the 4-bit setting, increasing the outlier ratio can further improve quality of GANQ, for example, with a 5% ratio, GANQ achieves a PPL of 6.25, which is comparable to [1] using a group size of 64.
>
> |WikiText PPL|3-bit
> |-|-|
> |[1] with group size 64|7.5
> |GANQ + 0.5%|7.46
>
> These results highlight a key trade-off between quality and memory, with our method offering a much more compact representation while maintaining strong quality.
>
> We will include these comparisons in the revision to highlight our method's unique benefits and compatibility with existing LUT-based kernels.
>
> ## Response to Weakness 1
>
> Thank you for raising the concern about real-time inference improvements. To clarify, we included inference benchmarks in Table 5. Following prior works [3,4], we used batch size 1 to generate 1024 tokens and reported total CUDA time (in seconds) using the LUT kernel from [3], as detailed in Section 4.1. The results show clear overall inference speedup and peak memory reduction for OPT-6.7B and LLaMA-7B, highlighting the practical benefits of our method over uniform quantization and FP16 baselines.
>
> We acknowledge the importance of testing across different batch sizes and sequence lengths and will expand our experiments accordingly. However, we emphasize that efficiency is largely determined by the underlying LUT-based kernel implementation.
>
> As noted in our response regarding LUT kernels [1,2], our main contribution is a novel LUT-based non-uniform quantization scheme. We plan to integrate optimized LUT kernels in future work to further improve efficiency. We appreciate the suggestion and will highlight the practical benefits more clearly. We believe that both our work and LUT kernel innovations can jointly make valuable contributions to the community.
>
> [3] Kim, Sehoon, et al. "Squeezellm: Dense-and-sparse quantization."
>
> [4] Frantar, Elias, et al. "Gptq: Accurate post-training quantization for generative pre-trained transformers."
>
> ## Response to Weakness 3
>
> We fully agree that open-sourcing will enhance reproducibility and impact. We will release the complete code upon acceptance.
>
> ## Typo
> Thank you for pointing out the typo, we will correct it in the revised version.

---

> > ### Comment · Reviewer_WtKA · 2025-04-04
> >
> > Thanks for the clarification and for sharing the additional results. However, I would like to see some comprehensive latency and throughput results under different batch sizes and sequence lengths.

---

> > > ### Author Response · Authors · 2025-04-05
> > >
> > > Thank you for your feedback. Below, we provide comprehensive latency speedup results for GANQ (with the kernel from [1]) relative to the FP16 model across various batch sizes and generation lengths on a single NVIDIA RTX 4090 GPU. Note that we slightly modified the benchmarking code to support batch sizes $>$ 1, resulting in minor differences from previous results.
> > >
> > > LLaMA-7b (4-bit):
> > >
> > > |Length / Batch Size|1|2|4|
> > > |:-:|-|-|-|
> > > |64|2.14|1.71|1.27|
> > > |128|2.12|1.70|1.26|
> > > |256|2.10|1.69|1.26|
> > > |512|2.06|1.66|1.26
> > > |1024|2.02|1.63|1.25
> > >
> > > LLaMA-7b (3-bit):
> > >
> > > |Length / Batch Size|1|2|4|
> > > |:-:|-|-|-|
> > > |64|2.45|1.95|1.39
> > > |128|2.40|1.91|1.38
> > > |256|2.39|1.90|1.36
> > > |512|2.33|1.85|1.35
> > > |1024|2.27|1.78|1.33
> > >
> > > As shown, speedup decreases with larger batch sizes or longer sequence lengths. This occurs because model compression primarily benefits memory efficiency, while increased batch sizes and sequence lengths shift the bottleneck toward computation, reducing memory-related gains. Reviewers QPHs and V6cW also highlighted this common phenomenon in weight quantization methods; we will explicitly include this analysis in the revised manuscript.
> > >
> > > We would also like to emphasize again that our primary contribution lies in **the accuracy improvement of LUT-based quantization** rather than kernel-level optimizations. Although our current experiments utilize the kernel from [1], our approach is compatible with future, potentially more optimized LUT-based kernels.
> > >
> > > Furthermore, we plan to revise Table 5 in the paper to explicitly highlight GANQ’s compatibility with existing LUT-based kernels and emphasize that our focus remains on more accurate quantization rather than kernel optimization.
> > >
> > > [1] Kim, Sehoon, et al. "Squeezellm: Dense-and-sparse quantization."

---

### Decision · Program_Chairs · 2025-05-01

**Decision:**

Accept (poster)

**Comment:**

This paper introduces GANQ, a GPU-adaptive non-uniform quantization framework for large language models, addressing key limitations in existing quantization approaches. GANQ reformulates quantization as a layer-wise mixed-integer optimization problem optimized for LUT-based inference, yielding impressive perplexity improvements and up to 2.57× speedup over FP16 baselines. While reviewers initially raised concerns regarding comparisons to related LUT methods, evaluation breadth, and profiling consistency, the authors provided detailed responses, expanded benchmarks, and clarified the impact of kernel implementations and experimental setups.
One remaining concern is that GANQ's efficiency gains, while solid, fall short of those reported for other 4-bit methods under optimized kernel settings, highlighting room for improvement in kernel-level integration. Given its empirical results and the authors’ thorough rebuttal and transparency, I recommend acceptance.